# Pruning Randomly Initialized Neural Networks with Iterative Randomization

**Daiki Chijiwa**[†][*]  **Shin'ya Yamaguchi**[†]  **Yasutoshi Ida**[†]

**Kenji Umakoshi**[‡]  **Tomohiro Inoue**[‡]

[†]NTT Computer and Data Science Laboratories, NTT Corporation
[‡]NTT Social Informatics Laboratories, NTT Corporation

## Abstract

Pruning the weights of randomly initialized neural networks plays an important role in the context of lottery ticket hypothesis. Ramanujan et al. [23] empirically showed that only pruning the weights can achieve remarkable performance instead of optimizing the weight values. However, to achieve the same level of performance as the weight optimization, the pruning approach requires more parameters in the networks before pruning and thus more memory space. To overcome this parameter inefficiency, we introduce a novel framework to prune randomly initialized neural networks with iteratively randomizing weight values (IteRand). Theoretically, we prove an approximation theorem in our framework, which indicates that the randomizing operations are provably effective to reduce the required number of the parameters. We also empirically demonstrate the parameter efficiency in multiple experiments on CIFAR-10 and ImageNet. The code is available at https://github.com/dchiji-ntt/iterand.

## 1  Introduction

The lottery ticket hypothesis, which was originally proposed by Frankle and Carbin [5], has been an important topic in the research of deep neural networks (DNNs). The hypothesis claims that an over-parameterized DNN has a sparse subnetwork (called a *winning ticket*) that can achieve almost the same accuracy as the fully-trained entire network when trained independently. If the hypothesis holds for a given network, then we can reduce the computational cost by using the sparse subnetwork instead of the entire network while maintaining the accuracy [14, 19]. In addition to the practical benefit, the hypothesis also suggests that the over-parametrization of DNNs is no longer necessary and their subnetworks alone are sufficient to achieve full accuracy.

Ramanujan et al. [23] went one step further. They proposed and empirically demonstrated a conjecture related to the above hypothesis, called the *strong lottery ticket hypothesis*, which informally states that there exists a subnetwork in a randomly initialized neural network such that it already achieves almost the same accuracy as a fully trained network, *without* any optimization of the weights of the network. A remarkable consequence of this hypothesis is that neural networks could be trained by solving a discrete optimization problem. That is, we may train a randomly initialized neural network by finding an optimal subnetwork (which we call *weight-pruning optimization*), instead of optimizing the network weights continuously (which we call *weight-optimization*) with stochastic gradient descent (SGD).

However, the weight-pruning optimization requires a problematic amount of parameters in the random network before pruning. Pensia et al. [22] theoretically showed that the required network width for the weight-pruning optimization needs to be logarithmically wider than the weight-optimization at

---

[*]Corresponding author: daiki.chijiwa.mk@hco.ntt.co.jp

least in the case of shallow networks. Therefore, the weight-pruning optimization requires more parameters, and thus more memory space, than the weight-optimization to achieve the same accuracy. In other words, under a given memory constraint, the weight-pruning optimization can have lower final accuracy than the weight-optimization in practice.

In this paper, we propose a novel optimization method for neural networks called *weight-pruning with iterative randomization* (IteRand), which extends the weight-pruning optimization to overcome the parameter inefficiency. The key idea is to virtually increase the network width by randomizing pruned weights at each iteration of the weight-pruning optimization, without any additional memory consumption. Indeed, we theoretically show that the required network width can be reduced by the randomizing operations. More precisely, our theoretical result indicates that, if the number of randomizing operations is large enough, we can reduce the required network width for weight-pruning to the same as that for a network fully trained by the weight-optimization up to constant factors, in contrast to the logarithmic factors of the previous results [22, 21]. We also empirically demonstrate that, under a given amount of network parameters, IteRand boosts the accuracy of the weight-pruning optimization in multiple vision experiments.

## 2 Background

In this section, we review the prior works on pruning randomly initialized neural networks.

**Notation and setup.** Let $d, N \in \mathbb{N}$. Let $f(x; \boldsymbol{\theta})$ be an $l$-layered ReLU neural network with an input $x \in \mathbb{R}^d$ and parameters $\boldsymbol{\theta} = (\theta_i)_{1 \le i \le n} \in \mathbb{R}^n$, where each weight $\theta_i$ is randomly sampled from a distribution $\mathcal{D}_{\text{param}}$ over $\mathbb{R}$. A subnetwork of $f(x; \boldsymbol{\theta})$ is written as $f(x; \boldsymbol{\theta} \odot \mathbf{m})$ where $\mathbf{m} \in \{0, 1\}^n$ is a binary mask and "$\odot$" represents an element-wise multiplication.

Ramanujan et al. [23] empirically observed that we can train the randomly initialized neural network $f(x; \boldsymbol{\theta})$ by solving the following discrete optimization problem, which we call *weight-pruning optimization*:

$$\min_{\mathbf{m} \in \{0,1\}^n} \mathbb{E}_{(x,y) \sim \mathcal{D}_{\text{labeled}}} \Big[ \mathcal{L}(f(x; \boldsymbol{\theta} \odot \mathbf{m}), y) \Big], \tag{1}$$

where $\mathcal{D}_{\text{labeled}}$ is a distribution on a set of labeled data $(x, y)$ and $\mathcal{L}$ is a loss function. To solve this optimization problem, Ramanujan et al. [23] proposed an optimization algorithm, called *edge-popup* (Algorithm 1).

---

**Algorithm 1:** Weight-pruning optimization by edge-popup [23]

1 Initialize $\boldsymbol{\theta} \sim \mathcal{D}_{\text{param}}^n, \mathbf{s} \sim \mathcal{D}_{\text{score}}^n$;     // $\mathcal{D}_{\text{param}}$ and $\mathcal{D}_{\text{score}}$ are distributions over $\mathbb{R}$
2 **while** $k = 0, \cdots, N - 1$ **do**
3    Sample a labeled data $(x, y) \sim \mathcal{D}_{\text{labeled}}$;
4    $\mathbf{m}, \mathbf{s} \leftarrow \texttt{TrainMask}(\boldsymbol{\theta}, \mathbf{s}; (x, y))$;     // optimize importance scores s and update m
5 **end**
6 return $\mathbf{m}, \boldsymbol{\theta}$;

---

The `TrainMask` (Algorithm 2) is the key process in Algorithm 1. It has a latent variable $\mathbf{s} = (s_i)_{1 \le i \le n} \in \mathbb{R}^n$, where each element $s_i$ represents an importance score of the corresponding weight $\theta_i$, and optimizes $\mathbf{s}$ instead of directly optimizing the discrete variable $\mathbf{m}$. Given the score $\mathbf{s}$, the corresponding $\mathbf{m}$ is computed by the function `CalculateMask(s)`, which returns $\mathbf{m} = (m_i)_{1 \le i \le n}$ defined as follows: $m_i = 1$ if $s_i$ is top $100(1 - p)\%$ in $\{s_i\}_{1 \le i \le n}$, otherwise $m_i = 0$, where $p \in (0, 1)$ is a hyperparameter representing a sparsity rate of the pruned network. In the line 3 of Algorithm 2, $\text{SGD}_{\eta, \lambda, \mu}(\mathbf{s}, \mathbf{g})$ returns the updated value of $\mathbf{s}$ by stochastic gradient descent with a learning rate $\eta$, weight decay $\lambda$, momentum coefficient $\mu$, and gradient vector $\mathbf{g}$.

On the theoretical side, Malach et al. [16] first provided a mathematical justification of the above empirical observation. They formulated it as an approximation theorem with some assumptions on the network width as follows.

**Theorem 2.1 (informal statement of Theorem 2.1 in [16])** *Let $f_{\text{target}}(x)$ be an l-layered network with bounded weight matrices, and $g(x)$ be a randomly initialized 2l-layered neural network. If the width of $g(x)$ is larger than $f_{\text{target}}(x)$ by the factor of a polynomial term, then there probably exists a subnetwork of $g(x)$ that approximates $f_{\text{target}}(x)$.*

---

**Algorithm 2:** Pseudo code of `TrainMask`

---

1  **Input**: $\boldsymbol{\theta}, \mathbf{s} \in \mathbb{R}^n$, $(x, y)$: a labeled data;
2  $\mathbf{m} \leftarrow \texttt{CalculateMask}(\mathbf{s})$;         `// calculate the mask m with the current scores`
3  $\mathbf{s} \leftarrow \text{SGD}_{\eta, \lambda, \mu}\left(\mathbf{s}, \nabla_{\bar{\mathbf{s}} = \mathbf{m}} \mathcal{L}(f(x; \boldsymbol{\theta} \odot \bar{\mathbf{s}}), y)\right)$;     `// update s by the gradient at` $\bar{\mathbf{s}} = \mathbf{m}$
4  $\mathbf{m} \leftarrow \texttt{CalculateMask}(\mathbf{s})$;         `// calculate new mask m with the updated scores`
5  return $\mathbf{m}, \mathbf{s}$;

---

By considering a well-trained network as $f_{\text{target}}$, Theorem 2.1 indicates that pruning a sufficiently wide $g(x)$ may reveal a subnetwork which achieves good test accuracy as $f_{\text{target}}$, in principle.

In the follow-up works [22, 21], the assumption on the network width was improved by reducing the factor of the required width to a logarithmic term. However, Pensia et al. [22] showed that the logarithmic order is unavoidable at least in the case of $l = 1$. While their results imply the optimality of the logarithmic bound, it also means that we cannot further relax the assumption on the network width as long as we work in the same setting. This indicates a limitation of the weight-pruning optimization, i.e. the weight-pruning optimization can train only less expressive models than ones trained with the conventional weight-optimization like SGD, under a given amount of memory or network parameters.

## 3  Method

In this section, we present a novel method called *weight-pruning with iterative randomization* (IteRand) for randomly initialized neural networks.

As discussed in Section 2, although the original weight-pruning optimization (Algorithm 1) can achieve good accuracy, it still has a limitation in the expressive power under a fixed amount of memory or network parameters. Our method is designed to overcome this limitation. The main idea is to randomize pruned weights at each iteration of the weight-pruning optimization. As we prove in Section 4, this reduces the required size of an entire network to be pruned.

We use the same notation and setup as Section 2. In addition, we assume that each weight $\theta_i$ of the network $f(x; \boldsymbol{\theta})$ can be re-sampled from $\mathcal{D}_{\text{param}}$ at each iteration of the weight-pruning optimization.

### 3.1  Algorithm

Algorithm 3 describes our proposed method, IteRand, which extends Algorithm 1. The differences from Algorithm 1 are lines 5-7 in Algorithm 3. IteRand has a hyperparameter $K_{\text{per}} \in \mathbb{N}_{\geq 1}$ (line 5). At the $k$-th iteration, whenever $k$ can be divided by $K_{\text{per}}$, pruned weights are randomized by `Randomize`$(\boldsymbol{\theta}, \mathbf{m})$ function (line 6). There are multiple possible designs for `Randomize`$(\boldsymbol{\theta}, \mathbf{m})$, which will be discussed in the next subsection.

---

**Algorithm 3:** Weight-pruning optimization with iterative randomization (IteRand)

---

1  Initialize $\boldsymbol{\theta} \sim \mathcal{D}_{\text{param}}, \mathbf{s} \sim \mathcal{D}_{\text{score}}$;
2  **while** $k = 0, \cdots, N - 1$ **do**
3      Sample a labeled data $(x, y) \sim \mathcal{D}_{\text{labeled}}$;
4      $\mathbf{m}, \mathbf{s} \leftarrow \texttt{TrainMask}(\boldsymbol{\theta}, \mathbf{s}; (x, y))$;
5      **if** $k + 1$ can be divided by $K_{\text{per}}$ **then** `// this if-block is newly added to Algorithm 1`
6         |  $\boldsymbol{\theta} \leftarrow \texttt{Randomize}(\boldsymbol{\theta}, \mathbf{m})$;         `// randomize a subset of pruned weights`
7      **end**
8  **end**
9  return $\mathbf{m}, \boldsymbol{\theta}$;

---

Note that $K_{\text{per}}$ controls how frequently the algorithm randomizes the pruned weights. Indeed the total number of the randomizing operations is $\lfloor N / K_{\text{per}} \rfloor$. If $K_{\text{per}}$ is too small, the algorithm is likely to be unstable because it may randomize even the important weights before their scores are well-optimized, and also the overhead of the randomizing operations cannot be ignored. In contrast,

if $K_{\mathrm{per}}$ is too large, the algorithm becomes almost same as the original weight-pruning Algorithm 1, and thus the effect of the randomization disappears. We fix $K_{\mathrm{per}} = 300$ on CIFAR-10 (about 1 epoch) and $K_{\mathrm{per}} = 1000$ on ImageNet (about $1/10$ epochs) in our experiments (Section 5).

### 3.2 Designs of `Randomize`$(\boldsymbol{\theta}, \mathbf{m})$

Here, we discuss how to define `Randomize`$(\boldsymbol{\theta}, \mathbf{m})$ function. There are several possible ways to randomize a subset of the parameters $\boldsymbol{\theta}$.

**Naive randomization.** For any distribution $\mathcal{D}_{\mathrm{param}}$, a naive definition of the randomization function (which we call *naive randomization*) can be given as follows.

$$\texttt{Randomize}(\boldsymbol{\theta}, \mathbf{m})_i := \begin{cases} \theta_i, & (\text{if } m_i = 1) \\ \widetilde{\theta}_i, & (\text{otherwise}) \end{cases}$$

where we denote the $i$-th component of `Randomize`$(\boldsymbol{\theta}, \mathbf{m}) \in \mathbb{R}^n$ as `Randomize`$(\boldsymbol{\theta}, \mathbf{m})_i$, and each $\widetilde{\theta}_i \in \mathbb{R}$ is a random variable with the distribution $\mathcal{D}_{\mathrm{param}}$. Also this can be written in another form as

$$\texttt{Randomize}(\boldsymbol{\theta}, \mathbf{m}) := \boldsymbol{\theta} \odot \mathbf{m} + \widetilde{\boldsymbol{\theta}} \odot (1 - \mathbf{m}), \tag{2}$$

where $\widetilde{\boldsymbol{\theta}} = (\widetilde{\theta}_i)_{1 \leq i \leq n} \in \mathbb{R}^n$ is a random variable with the distribution $\mathcal{D}_{\mathrm{param}}^n$.

**Partial randomization.** The naive randomization (Eq. (2)) is likely to be unstable because it entirely replaces all pruned weights with random values every $K_{\mathrm{per}}$ iteration. To increase the stability, we modify the definition of the naive randomization as it replaces a randomly chosen subset of the pruned weights as follows (which we call *partial randomization*):

$$\texttt{Randomize}(\boldsymbol{\theta}, \mathbf{m}) := \boldsymbol{\theta} \odot \mathbf{m} + \left( \boldsymbol{\theta} \odot (1 - \mathbf{b}_r) + \widetilde{\boldsymbol{\theta}} \odot \mathbf{b}_r \right) \odot (1 - \mathbf{m}), \tag{3}$$

where $\widetilde{\boldsymbol{\theta}} \in \mathbb{R}^n$ is the same as in Eq. (2), $r \in [0, 1]$ is a hyperparameter and $\mathbf{b}_r = (b_{r,i})_{1 \leq i \leq n} \in \{0, 1\}^n$ is a binary vector whose each element is sampled from the Bernoulli distribution $\mathrm{Bernoulli}(r)$, i.e. $b_{r,i} = 1$ with probability $r$ and $b_{r,i} = 0$ with probability $1 - r$.

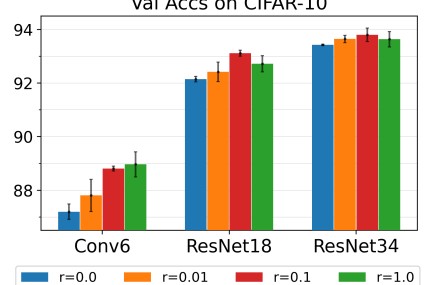

The partial randomization replaces randomly chosen $100r\%$ of all pruned weights with random values. Note that, when $r = 1$, the partial randomization is equivalent to the naive randomization (Eq. (2)). In contrast, when $r = 0$, it never randomizes any weights and thus is equivalent to Algorithm 1.

In Figure 1, we observe that $r = 0.1$ works well with various network architectures on CIFAR-10, where we use the Kaiming uniform distribution (whose definition will be given in Section 5) for $\mathcal{D}_{\mathrm{param}}$ and $K_{\mathrm{per}} = 300$.

Figure 1: **Analysis on $r$.** We compare partial randomizations with $r \in \{0.0, 0.01, 0.1, 1.0\}$ applied to CNNs. The y-axis is the validation accuracy on CIFAR-10. $r = 0.1$ achieves better mean accuracy for every CNNs.

## 4 Theoretical justification

In this section, we present a theoretical justification for our iterative randomization approach on the weight-pruning of randomly initialized neural networks.

### 4.1 Setup

We consider a target neural network $f : \mathbb{R}^{d_0} \to \mathbb{R}^{d_l}$ of depth $l$, which is described as follows.

$$f(x) = F_l \sigma(F_{l-1} \sigma(\cdots F_1(x) \cdots)), \tag{4}$$

where $x$ is a $d_0$-dimensional real vector, $\sigma$ is the ReLU activation, and $F_i$ is a $d_i \times d_{i-1}$ matrix. Our objective is to approximate the target network $f(x)$ by pruning a randomly initialized neural network $g(x)$, which tends to be larger than the target network.

Similar to the previous works [16, 22], we assume that $g(x)$ is twice as deep as the target network $f(x)$. Thus, $g(x)$ can be described as

$$g(x) = G_{2l}\sigma(G_{2l-1}\sigma(\cdots G_1(x)\cdots)), \tag{5}$$

where $G_j$ is a $\widetilde{d}_j \times \widetilde{d}_{j-1}$ matrix ($j = 1, \cdots, 2l$) with $\widetilde{d}_{2i} = d_i$. Each element of the matrix $G_j$ is assumed to be drawn from the uniform distribution $U[-1, 1]$. Since there is a one-to-one correspondence between pruned networks of $g(x)$ and sequences of binary matrices $M = \{M_j\}_{j=1,\cdots,2l}$ with $M_j \in \{0,1\}^{\widetilde{d}_j \times \widetilde{d}_{j-1}}$, every pruned network of $g(x)$ can be described as

$$g_M(x) = (G_{2l} \odot M_{2l})\sigma((G_{2l-1} \odot M_{2l-1})\sigma(\cdots(G_1 \odot M_1)(x)\cdots)). \tag{6}$$

Under these setups, we recall that the previous works showed that, with high probability, there exists a subnetwork of $g(x)$ that approximates $f(x)$ when the width of $g(x)$ is larger than $f(x)$ by polynomial factors [16] or logarithmic factors [22, 21].

## 4.2 Formulation and main results

Now we attempt to mathematically formulate our proposed method, IteRand, as an approximation problem. As described in Algorithm 3, the method consists of two steps: optimizing binary variables $M = \{M_j\}_{j=1,\cdots,2l}$ and randomizing pruned weights in $g(x)$. The first step can be formulated as the approximation problem of $f(x)$ by some $g_M(x)$ as described above. Corresponding to the second step, we introduce an idealized assumption on $g(x)$ for a given number $R \in \mathbb{N}_{\geq 1}$: each element of the weight matrix $G_j$ can be re-sampled with replacement from the uniform distribution $U[-1, 1]$ up to $R - 1$ times, for all $j = 1, \cdots, 2l$. (*re-sampling assumption for R*)

Under this re-sampling assumption, we obtain the following theorem.

**Theorem 4.1 (Main Theorem)** *Fix $\epsilon, \delta > 0$, and we assume that $\|F_i\|_{\mathrm{Frob}} \leq 1$. Let $R \in \mathbb{N}$, and assume that $g(x)$ satisfies the re-sampling assumption for $R$.*
*If $\widetilde{d}_{2i-1} \geq 2d_{i-1}\lceil \frac{64l^2 d_{i-1}^2 d_i}{\epsilon^2 R^2} \log(\frac{2ld_{i-1}d_i}{\delta})\rceil$ holds for all $i = 1, \cdots, l$, then with probability at least $1 - \delta$, there exist binary matrices $M = \{M_j\}_{1 \leq j \leq 2l}$ such that*

$$\|f(x) - g_M(x)\|_2 \leq \epsilon, \text{ for } \|x\|_\infty \leq 1. \tag{7}$$

*In particular, if $R$ is larger than $\frac{8ld_{i-1}}{\epsilon}\sqrt{d_i \log(\frac{2ld_{i-1}d_i}{\delta})}$, then $\widetilde{d}_{2i-1} = 2d_{i-1}$ is enough.*

Theorem 4.1 shows that the iterative randomization is provably helpful to approximate wider networks in the weight-pruning optimization of a random network. In fact, the required width for $g(x)$ in Theorem 4.1 is reduced to even twice as wide as $f(x)$ when the number of re-sampling is sufficiently large, in contrast to the prior results without re-sampling assumption where the required width is logarithmically wider than $f(x)$ [22, 21]. This means that, under a fixed amount of parameters of $g(x)$, we can achieve a higher accuracy by weight-pruning of $g(x)$ with iterative randomization since a wider target network has a higher model capacity.

In the rest of this section we present the core ideas by proving the simplest case ($l = d_0 = d_1 = 1$) of the theorem, while the full proof of Theorem 4.1 is given in Appendix A. Note that the full proof is obtained essentially by applying the argument for the simplest case inductively on the widths and depths.

## 4.3 Proof ideas for Theorem 4.1

Let us consider the case of $l = d_0 = d_1 = 1$. Then the target network $f(x)$ can be written as $f(x) = wx : \mathbb{R} \to \mathbb{R}$, where $w \in \mathbb{R}$, and $g(x)$ can be written as $g(x) = \mathbf{v}^T\sigma(\mathbf{u}x)$ where $\mathbf{u}, \mathbf{v} \in \mathbb{R}^{\widetilde{d}_1}$. Also, subnetworks of $g(x)$ can be written as $g_{\mathbf{m}}(x) = (\mathbf{v} \odot \mathbf{m})^T\sigma((\mathbf{u} \odot \mathbf{m})x)$ for some $\mathbf{m} \in \{0,1\}^{\widetilde{d}_1}$.

There are two technical points in our proof. The first point is the following splitting of $f(x)$:

$$f(x) = w\sigma(x) - w\sigma(-x), \tag{8}$$

for any $x \in \mathbb{R}$. This splitting is very similar to the one used in the previous works [16, 22, 21]:

$$f(x) = \sigma(wx) - \sigma(-wx). \tag{9}$$

However, if we use the latter splitting Eq. (9), it turns out that we cannot obtain the lower bound of $\widetilde{d}_{2i-1}$ in Theorem 4.1 when $d_0 > 0$. (Here we do not treat this case, but the proof for $d_0 > 0$ is given in Appendix A.) Thus we need to use our splitting Eq. (8) instead.

Using Eq. (8), we can give another proof of the following approximation result without iterative randomization, which was already shown in the previous work [16].

**Lemma 4.2** *Fix $\epsilon, \delta \in (0,1), w \in [-1,1], d \in \mathbb{N}$. Let $\mathbf{u}, \mathbf{v} \sim U[-1,1]^d$ be uniformly random weights of a 2-layered neural network $g(x) := \mathbf{v}^T \sigma(\mathbf{u} \cdot x)$. If $d \geq 2\lceil \frac{16}{\epsilon^2} \log(\frac{2}{\delta}) \rceil$ holds, then with probability at least $1 - \delta$,*

$$\big| wx - g_{\mathbf{m}}(x) \big| \leq \epsilon, \text{ for all } x \in \mathbb{R}, |x| \leq 1, \tag{10}$$

*where $g_{\mathbf{m}}(x) := (\mathbf{v} \odot \mathbf{m})^T \sigma(\mathbf{u} \cdot x)$ for some $\mathbf{m} \in \{0,1\}^d$.*

**Proof (sketch):** We assume that $d$ is an even number as $d = 2d'$ so that we can split an index set $\{0, \cdots, d-1\}$ of $d$ hidden neurons of $g(x)$ into $I = \{0, \cdots, d'-1\}$ and $J = \{d', \cdots, d-1\}$. Then we have the corresponding subnetworks $g_I(x)$ and $g_J(x)$ given by $g_I(x) := \sum_{k \in I} v_k \sigma(u_k x), g_J(x) := \sum_{k \in J} v_k \sigma(u_k x)$, which satisfy the equation $g(x) = g_I(x) + g_J(x)$.

By the splitting Eq. (8), it is enough to consider the probabilities for approximating $w\sigma(x)$ by a subnetwork of $g_I(x)$ and for approximating $-w\sigma(-x)$ by a subnetwork of $g_J(x)$. Now we have

$$\mathbb{P}\left( \nexists i \in I \text{ s.t. } |u_i - 1| \leq \frac{\epsilon}{2}, |v_i - w| \leq \frac{\epsilon}{2} \right) \leq \left( 1 - \frac{\epsilon^2}{16} \right)^{d'} \leq \frac{\delta}{2}, \tag{11}$$

$$\mathbb{P}\left( \nexists j \in J \text{ s.t. } |u_j + 1| \leq \frac{\epsilon}{2}, |v_j + w| \leq \frac{\epsilon}{2} \right) \leq \left( 1 - \frac{\epsilon^2}{16} \right)^{d'} \leq \frac{\delta}{2}, \tag{12}$$

for $d' \geq \frac{16}{\epsilon^2} \log \left( \frac{2}{\delta} \right)$, by a standard argument of the uniform distribution and the inequality $e^x \geq 1 + x$ for $x \geq 0$. By the union bound, with probability at least $1 - \delta$, we have $i \in I$ and $j \in J$ such that

$$\big| w\sigma(x) - v_i \sigma(u_i x) \big| \leq \frac{\epsilon}{2},$$

$$\big| -w\sigma(-x) - v_j \sigma(u_j x) \big| \leq \frac{\epsilon}{2}.$$

Combining these inequalities, we finish the proof. □

The second point of our proof is introducing projection maps to leverage the re-sampling assumption, as follows. As in the proof of Lemma 4.2, we assume that $d = 2d'$ for some $d' \in \mathbb{N}$ and let $I = \{0, \cdots, d'-1\}, J = \{d', \cdots, d-1\}$. Now we define a projection map

$$\pi : \widetilde{I} \to I, \quad k \mapsto \lfloor k/R \rfloor, \tag{13}$$

where $\widetilde{I} := \{0, \cdots, d'R - 1\}$, and $\lfloor \cdot \rfloor$ denotes the floor function. Similarly for $J$, we can define $\widetilde{J} := \{d'R, \cdots, dR - 1\}$ and the corresponding projection map. Using these projection maps, we can extend Lemma 4.2 to the one with the re-sampling assumption, which is the special case of Theorem 4.1:

**Proposition 4.3 (Theorem 4.1 with $l = d_0 = d_1 = 1$)** *Fix $\epsilon, \delta \in (0,1), w \in [-1,1], d \in \mathbb{N}$. Let $\mathbf{u}, \mathbf{v} \sim U[-1,1]^d$ be uniformly random weights of a 2-layered neural network $g(x) := \mathbf{v}^T \sigma(\mathbf{u} \cdot x)$. Let $R \in \mathbb{N}$ and we assume that each element of $\mathbf{u}$ and $\mathbf{v}$ can be re-sampled with replacement up to $R - 1$ times. If $d \geq 2\lceil \frac{16}{\epsilon^2 R^2} \log(\frac{2}{\delta}) \rceil$ holds, then with probability at least $1 - \delta$,*

$$\big| wx - g_{\mathbf{m}}(x) \big| \leq \epsilon, \text{ for all } x \in \mathbb{R}, |x| \leq 1, \tag{14}$$

*where $g_{\mathbf{m}}(x) := (\mathbf{v} \odot \mathbf{m})^T \sigma(\mathbf{u} \cdot x)$ for some $\mathbf{m} \in \{0,1\}^d$.*

**Proof (sketch):** Similarly to the proof of Lemma 4.2, we utilize the splitting Eq. (8). We mainly argue on the approximation of $w\sigma(x)$ since the argument for approximating $-w\sigma(x)$ is parallel.

By the assumption that each element of $\mathbf{u}$ and $\mathbf{v}$ can be re-sampled up to $R-1$ times, we can replace the probability in Eq. (11) in the proof of Lemma 4.2, using the projection map $\pi : \widetilde{I} \to I$, by

$$\mathbb{P}\left(\nexists i_1, i_2 \in \widetilde{I} \text{ s.t. } \pi(i_1) = \pi(i_2), \ |\widetilde{u}_{i_1} - 1| \le \frac{\epsilon}{2}, \ |\widetilde{v}_{i_2} - w| \le \frac{\epsilon}{2}\right), \tag{15}$$

where $\widetilde{u}_1, \cdots, \widetilde{u}_{d'R}, \widetilde{v}_1, \cdots, \widetilde{v}_{d'R} \sim U[-1,1]$. Indeed, since we have

$$\#\{(i_1, i_2) \in \widetilde{I} \times \widetilde{I} : \pi(i_1) = \pi(i_2)\} = d'R^2, \tag{16}$$

we can evaluate the probability Eq. (15) as

$$\text{Eq.}(15) \le \left(1 - \frac{\epsilon^2}{16}\right)^{d'R^2} \le \frac{\delta}{2}, \tag{17}$$

for $d' \ge \frac{16}{\epsilon^2 R^2} \log\left(\frac{\delta}{2}\right)$. Eq. (17) can play the same role as Eq. (11) in the proof of Lemma 4.2.

Parallel argument can be applied for the approximation of $-w\sigma(x)$ by replacing $I$ with $J$. The rest of the proof is the same as Lemma 4.2. $\qquad\square$

## 5 Experiments

In this section, we perform several experiments to evaluate our proposed method, IteRand (Algorithm 3). Our main aim is to empirically verify the parameter efficiency of IteRand, compared with edge-popup [23] (Algorithm 1) on which IteRand is based. Specifically, we demonstrate that IteRand can achieve better accuracy than edge-popup under a given amount of network parameters. In all experiments, we used the partial randomization with $r = 0.1$ (Eq. (3)) for `Randomize` in Algorithm 3.

**Setup.** We used two vision datasets: CIFAR-10 [12] and ImageNet [25]. CIFAR-10 is a small-scale dataset of $32 \times 32$ images with 10 class labels. It has 50k images for training and 10k for testing. We randomly split the 50k training images into 45k for actual training and 5k for validation. ImageNet is a dataset of $224 \times 224$ images with 1000 class labels. It has the train set of $1.28$ million images and the validation set of 50k images. We randomly split the training images into $99 : 1$, and used the former for actual training and the latter for validating models. When testing models, we used the validation set of ImageNet (which we refer to as the test set). For network architectures, we used multiple convolutional neural networks (CNNs): Conv6 [5] as a shallow network and ResNets [10] as deep networks. Conv6 is a 6-layered VGG-like CNN, which is also used in the prior work [23]. ResNets are more practical CNNs with skip connections and batch normalization layers. Following the settings in Ramanujan et al. [23], we used non-affine batch normalization layers, which are layers that only normalize their inputs and do not apply any affine transform, when training ResNets with edge-popup and IteRand. All of our experiments were performed with 1 GPU (NVIDIA GTX 1080 Ti, 11GB) for CIFAR-10 and 2 GPUs (NVIDIA V100, 16GB) for ImageNet. The details of the network architectures and hyperparameters for training are given in Appendix B.

**Parameter distributions.** With the same notation as Section 2, both IteRand and edge-popup requires two distributions: $\mathcal{D}_{\text{param}}$ and $\mathcal{D}_{\text{score}}$. In our experiments, we consider Kaiming uniform (KU) and signed Kaiming constant (SC) distribution. The KU distribution is the uniform distribution over the interval $[-\sqrt{\frac{6}{c_{\text{fanin}}}}, \sqrt{\frac{6}{c_{\text{fanin}}}}]$ where $c_{\text{fanin}}$ is the constant defined for each layer of ReLU neural networks [1][9]. The SC distribution is the uniform distribution over the two-valued set $\{-\sqrt{\frac{2}{c_{\text{fanin}}}}, \sqrt{\frac{2}{c_{\text{fanin}}}}\}$, which is introduced by Ramanujan et al. [23]. We fix $\mathcal{D}_{\text{score}}$ to the KU distribution, and use the KU or SC distribution for $\mathcal{D}_{\text{param}}$.

### 5.1 Varying the network width

To demonstrate the parameter efficiency, we introduce a hyperparameter $\rho$ of the width factor for Conv6, ResNet18 and ResNet34. The details of this modification are given in Appendix B. We train and test these networks on CIFAR-10 using IteRand, edge-popup and SGD, varying the width factor $\rho$ in $\{0.25, 0.5, 1.0, 2.0\}$ for each network (Figure 2). In the experiments for IteRand and edge-popup, we used the sparsity rate of $p = 0.5$ for Conv6 and $p = 0.6$ for ResNet18 and ResNet34. Our method outperforms the baseline method for various widths. The difference in accuracy is large both when the width is small ($\rho \le 0.5$), and when $\mathcal{D}_{\text{param}}$ is the KU distribution where edge-popup struggles.

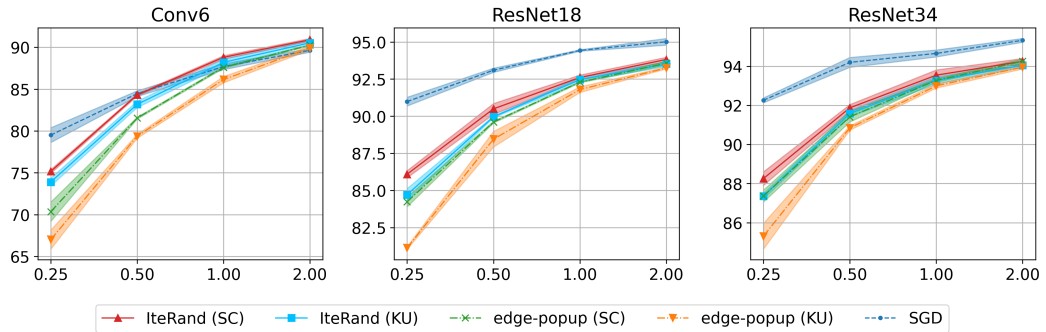

Figure 2: We train and evaluate three CNNs (Conv6 , ResNet18 and ResNet34) with various widths on CIFAR-10. The x-axis is the width factor $\rho$ and the y-axis is the accuracy on the test set. We plot the mean $\pm$ standard deviation over three runs for each experiment. KU/SC in the legend represents the distribution used for $\mathcal{D}_{\mathrm{param}}$.

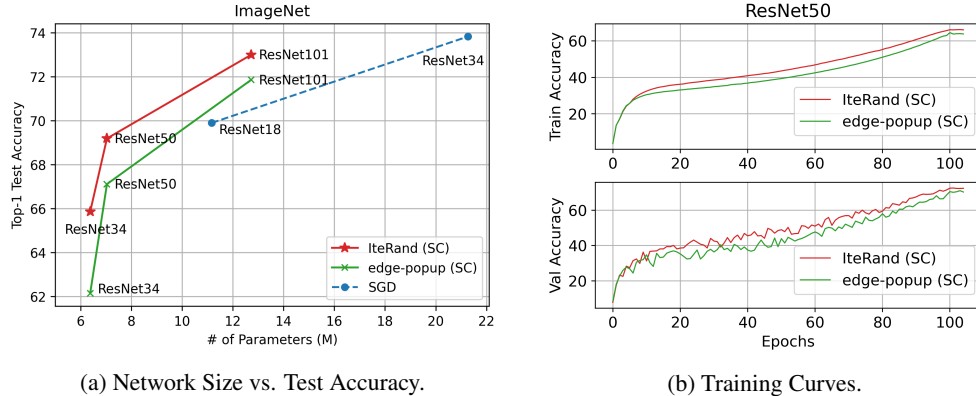

(a) Network Size vs. Test Accuracy.

(b) Training Curves.

Figure 3: We compare IteRand with edge-popup on ImageNet. We used a fixed $p = 0.7$ for the sparsity rate (Section 2) following Ramanujan et al. [23]. Figure (a) shows parameter-accuracy tradeoff curves on the test set. In Figure (b), we plot the train/val accuracy of ResNet50 during optimization. IteRand outperforms edge-popup from early epochs.

## 5.2 ImageNet experiments

The parameter efficiency of our method is also confirmed on ImageNet (Figure 3a). ImageNet is more difficult than CIFAR-10, and thus more complexity is required for networks to achieve competitive performance. Since our method can increase the network complexity as shown in Section 4, the effect is significant in ImageNet especially when the complexity is limited such as ResNet34.

In addition to the parameter efficiency, we also observe the effect of iterative randomization on the behavior of optimization process, by plotting training curves (Figure 3b). Surprisingly, IteRand achieves significantly better performance than edge-popup at the early stage of the optimization, which indicates that the iterative randomization accelerates the optimization process especially when the number of iterations is limited.

## 6 Related work

**Lottery ticket hypothesis**   Frankle and Carbin [5] originally proposed the lottery ticket hypothesis. Many properties and applications have been studied in subsequent works, such as transferability of winning subnetworks [19], sparsification before training [14, 27, 26] and further ablation studies on the hypothesis [29].

Zhou et al [29] surprisingly showed that only pruning randomly initialized networks without any optimization on their weights can be a training method surprisingly. Ramanujan et al. [23] went

further by proposing an effective pruning algorithm on random networks, called edge-popup, and achieved competitive accuracy compared with standard training algorithms by weight-optimization [24]. Malach et al. [16] mathematically formalized their pruning-only approach as an approximation problem for a target network and proved it with lower bound condition on the width of random networks to be pruned. Subsequent works [22, 21] successfully relaxed the lower bound to the logarithmic factor wider than the width of a target network. Our work can be seen as an extension of their works to allow re-sampling of the weights of the random networks for finite $R$ times, and we showed that the logarithmic factor can be reduced to a constant one when $R$ is large enough. (See Section 4.)

**Neural network pruning and regrowth**    Studies of finding sparse structures of neural networks date back to the late 1980s [8, 13]. There are many approaches to sparsify networks, such as magnitude-based pruning [7], $L_0$ regularization [15] and variational dropout [18]. Although these methods only focus on pruning unnecessary weights from the networks, there are several studies on re-adding new weights during sparsification [11] to maintain the model complexity of the sparse networks. Han et al. [6] proposed a dense-sparse-dense (DSD) training algorithm consisting of three stages: dense training, pruning, and recovering the pruned weights as zeros followed by dense training. Mocanu et al. [17] proposed sparse evolutionary training (SET), which repeats pruning and regrowth with random values at the end of each training epoch. Pruning algorithms proposed in other works [2, 20, 4] are designed to recover pruned weights by zero-initialization instead of random values, so that the recovered weights do not affect the outputs of the networks. While these methods are similar to our iterative randomization method in terms of the re-adding processes, all of them use weight-optimization to train networks including re-added weights, in contrast to our pruning-only approach.

## 7    Limitations

There are several limitations in our theoretical results. (1) Theorem 4.1 indicates only the existence of subnetworks that approximate a given neural network, not whether our method works well empirically. (2) The theorem focused on the case when the parameter distribution $\mathcal{D}_{\mathrm{param}}$ is the uniform distribution over the interval $[-1, 1]$. Thus, generalizing our theorem to other distributions, such as the uniform distribution over binary values $\{-1, 1\}$ [3], is left for future work. (3) The required width given in the theorem may not be optimal. Indeed, prior work [22] showed that we can reduce the polynomial factors in the required width to logarithmic ones in the case when the number of re-sampling operations $R = 1$. Whether we can reduce the required width for $R > 1$ remains an open question.

Also our algorithm (IteRand) and its empirical results have several limitations. (1) Pruning randomly-initialized networks without any randomization can reduce the storage cost by saving only a single random seed and the binary mask representing the optimal subnetwork. However, if we save the network pruned with IteRand in the same way, it requires more storage cost: $R$ random seeds and $R$ binary masks, where $R$ is the number of re-samplings. (2) Although our method can be applied with any score-based pruning algorithms (e.g. Zhou et al [29] and Wang et al [28]), we evaluated our method only combined with edge-popup [23], which is the state-of-the-art algorithm for pruning random networks. Since our theoretical results do not depend on any pruning algorithms, we expect that our method can be effectively combined with better pruning algorithms that emerge in the future. (3) We performed experiments mainly on the tasks of image classification. An intriguing question is how effectively our method works on other tasks such as language understanding, audio recognition, and deep reinforcement learning with various network architectures.

## 8    Conclusion

In this paper, we proposed a novel framework of iterative randomization (IteRand) for pruning randomly initialized neural networks. IteRand can virtually increase the network widths without any additional memory consumption, by randomizing pruned weights of the networks iteratively during the pruning procedure. We verified its parameter efficiency both theoretically and empirically.

Our results indicate that the weight-pruning of random networks may become a practical approach to train the networks when we apply the randomizing operations enough times. This opens up the

possibility that the weight-pruning can be used instead of the standard weight-optimization within the same memory budget.

## Acknowledgement

The authors thank Kyosuke Nishida and Hengjin Tang for valuable discussions in the early phase of our study. We also thank Osamu Saisho for helpful comments on the manuscript. We appreciate CCI team in NTT for building and maintaining their computational cluster on which most of our experiments were computed.

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
