# Appendices for "Pruning Randomly Initialized Neural Networks with Iterative Randomization"

## Contents

## A    Proof of main theorem (Theorem 4.1)

### A.1    Settings and main theorem

Let $d_0, \cdots, d_l \in \mathbb{N}_{\geq 1}$. We consider a target neural network $f : \mathbb{R}^{d_0} \to \mathbb{R}^{d_l}$ of depth $l$, which is described as follows.

$$f(x) = F_l\sigma(F_{l-1}\sigma(\cdots F_1(x)\cdots)), \tag{1}$$

where $x$ is a $d_0$-dimensional real vector, $\sigma$ is the ReLU activation, and $F_i$ is a $d_i \times d_{i-1}$ matrix. Our objective is to approximate the target network $f(x)$ by pruning a randomly initialized neural network $g(x)$, which tends to be larger than the target network.

Similar to the previous works [6, 7], we assume that $g(x)$ is twice as deep as the target network $f(x)$. Thus, $g(x)$ can be described as

$$g(x) = G_{2l}\sigma(G_{2l-1}\sigma(\cdots G_1(x)\cdots)), \tag{2}$$

where $G_j$ is a $\widetilde{d}_j \times \widetilde{d}_{j-1}$ matrix ($\widetilde{d}_j \in \mathbb{N}_{\geq 1}$ for $j = 1, \cdots, 2l$) with $\widetilde{d}_{2i} = d_i$. Each element of the matrix $G_j$ is assumed to be drawn from the uniform distribution $U[-1, 1]$. Since there is a one-to-one correspondence between pruned networks of $g(x)$ and sequences of binary matrices $M = \{M_j\}_{j=1,\cdots,2l}$ with $M_j \in \{0, 1\}^{\widetilde{d}_j \times \widetilde{d}_{j-1}}$, every pruned network of $g(x)$ can be described as

$$g_M(x) = (G_{2l} \odot M_{2l})\sigma((G_{2l-1} \odot M_{2l-1})\sigma(\cdots(G_1 \odot M_1)(x)\cdots)). \tag{3}$$

We introduce an idealized assumption on $g(x)$ for a given number $R \in \mathbb{N}_{\geq 1}$: each element of the weight matrix $G_j$ can be re-sampled with replacement from the uniform distribution $U[-1, 1]$ up to $R - 1$ times, for all $j = 1, \cdots, 2l$ (*re-sampling assumption for $R$*). Here, re-sampling with replacement means that we sample a new value for an element of $G_j$ and replace the old value of the element by the new one.

Under this re-sampling assumption, we describe our main theorem as follows.

**Theorem A.1 (Main Theorem)** *Fix $\epsilon, \delta > 0$, and we assume that $\|F_i\|_{\mathrm{Frob}} \leq 1$. Let $R \in \mathbb{N}$ and we assume that each element of $G_i$ can be re-sampled with replacement from the uniform distribution $U[-1, 1]$ up to $R - 1$ times.*

*If $\widetilde{d}_{2i-1} \geq 2d_{i-1} \lceil \frac{64 l^2 d_{i-1}^2 d_i}{\epsilon^2 R^2} \log(\frac{2l d_{i-1} d_i}{\delta}) \rceil$ holds for all $i = 1, \cdots, l$, then with probability at least $1 - \delta$, there exist binary matrices $M = \{M_j\}_{1 \leq j \leq 2l}$ such that*

$$\|f(x) - g_M(x)\|_2 \leq \epsilon, \text{ for } \|x\|_\infty \leq 1. \tag{4}$$

*In particular, if $R$ is larger than $\frac{8 l d_{i-1}}{\epsilon} \sqrt{d_i \log(\frac{2l d_{i-1} d_i}{\delta})}$, then $\widetilde{d}_{2i-1} = 2d_{i-1}$ is enough.*

## A.2 Proof of Theorem A.1

Our proof is based on the following simple observation, similar to the arguments in Malach et al. [6].

**Lemma A.2** *Fix some $n \in \mathbb{N}$, $\alpha \in [-1, 1]$ and $\epsilon, \delta \in (0, 1)$. Let $X_1, \cdots, X_n \sim U[-1, 1]$. If $n \geq \frac{2}{\epsilon} \log(\frac{1}{\delta})$ holds, then with probability at least $1 - \delta$, we have*

$$|\alpha - X_i| \leq \epsilon, \tag{5}$$

*for some $i \in \{1, \cdots, n\}$.*

**Proof:** We can assume $\alpha \geq 0$ without loss of generality. By considering half $\epsilon$-ball of $\alpha$, we have

$$\mathbb{P}_{X \sim U[-1,1]}\left[\left|\alpha - X\right| \leq \epsilon\right] \geq \frac{\epsilon}{2}.$$

Thus it follows that

$$\mathbb{P}_{X_1, \cdots, X_n \sim U[-1,1]}\left[\left|\alpha - X_i\right| > \epsilon \text{ for all } i\right] \leq (1 - \frac{\epsilon}{2})^n \leq e^{-\frac{n\epsilon}{2}} \leq \delta.$$

$\square$

First, we consider to approximate a single variable linear function $f(x) = wx : \mathbb{R} \to \mathbb{R}, w \in \mathbb{R}$ by some subnetwork of 2-layered neural network $g(x)$ with $d$ hidden neurons, *without* the re-sampling assumption. Note that this is the same setting as in Malach et al. [6], but we give another proof so that we can later extend it to the one with the resumpling assumption.

**Lemma A.3** *Fix $\epsilon, \delta \in (0, 1), w \in [-1, 1], d \in \mathbb{N}$. Let $\mathbf{u}, \mathbf{v} \sim U[-1, 1]^d$ be uniformly random weights of a 2-layered neural network $g(x) := \mathbf{v}^T \sigma(\mathbf{u} \cdot x)$. If $d \geq 2\lceil \frac{16}{\epsilon^2} \log(\frac{2}{\delta}) \rceil$ holds, then with probability at least $1 - \delta$,*

$$\left|wx - g_{\mathbf{m}}(x)\right| \leq \epsilon, \text{ for all } x \in \mathbb{R}, |x| \leq 1, \tag{6}$$

*where $g_{\mathbf{m}}(x) := (\mathbf{v} \odot \mathbf{m})^T \sigma(\mathbf{u} \cdot x)$ for some $\mathbf{m} \in \{0, 1\}^d$.*

**Proof:** The core idea is to decompose $wx$ as

$$wx = w(\sigma(x) - \sigma(-x)) = w\sigma(x) - w\sigma(-x). \tag{7}$$

We assume that $d$ is an even number as $d = 2d'$ so that we can split an index set $\{1, \cdots, d\}$ of hidden neurons of $g(x)$ into $I = \{1, \cdots, d'\}$ and $J = \{d' + 1, \cdots, d\}$. Then we have the corresponding subnetworks $g_I(x)$ and $g_J(x)$ given by $g_I(x) := \sum_{k \in I} v_k \sigma(u_k x), g_J(x) := \sum_{k \in J} v_k \sigma(u_k x)$, which satisfy the equation $g(x) = g_I(x) + g_J(x)$.

From Eq. (7), it is enough to consider the probabilities for approximating $w\sigma(x)$ by a subnetwork of $g_I(x)$ and for approximating $-w\sigma(-x)$ by a subnetwork of $g_J(x)$. Now we have

$$\mathbb{P}\left(\nexists i \in I \text{ s.t. } |u_i - 1| \leq \frac{\epsilon}{2}, |v_i - w| \leq \frac{\epsilon}{2}\right) \leq \left(1 - \frac{\epsilon^2}{16}\right)^{d'} \leq \frac{\delta}{2}, \tag{8}$$

$$\mathbb{P}\left(\nexists j \in J \text{ s.t. } |u_j + 1| \leq \frac{\epsilon}{2}, |v_j + w| \leq \frac{\epsilon}{2}\right) \leq \left(1 - \frac{\epsilon^2}{16}\right)^{d'} \leq \frac{\delta}{2}, \tag{9}$$

for $d' \geq \frac{16}{\epsilon^2} \log\left(\frac{2}{\delta}\right)$ as well as in the proof of Lemma A.2. By using the union bound, with probability at least $1 - \delta$, we have $i \in I$ and $j \in J$ such that

$$\left| w\sigma(x) - v_i\sigma(u_i x) \right| \leq \frac{\epsilon}{2},$$
$$\left| -w\sigma(-x) - v_j\sigma(u_j x) \right| \leq \frac{\epsilon}{2}.$$

Combining these inequalities and Eq. (7), we finish the proof. $\qquad\square$

Now we extend Lemma A.3 to the one with the re-sampling assumption.

**Lemma A.4** *Fix* $\epsilon, \delta \in (0,1), w \in [-1,1], d \in \mathbb{N}$. *Let* $\mathbf{u}, \mathbf{v} \sim U[-1,1]^d$ *be uniformly random weights of a 2-layered neural network* $g(x) := \mathbf{v}^T \sigma(\mathbf{u} \cdot x)$. *Let* $R \in \mathbb{N}$ *and we assume that each elements of* $u$ *and* $v$ *can be re-sampled with replacement up to* $R-1$ *times. If* $d \geq 2\lceil \frac{16}{\epsilon^2 R^2} \log(\frac{2}{\delta}) \rceil$ *holds, then with probability at least* $1 - \delta$,

$$\left| wx - g_{\mathbf{m}}(x) \right| \leq \epsilon, \text{ for all } x \in \mathbb{R}, |x| \leq 1, \tag{10}$$

*where* $g_{\mathbf{m}}(x) := (\mathbf{v} \odot \mathbf{m})^T \sigma(\mathbf{u} \cdot x)$ *for some* $\mathbf{m} \in \{0,1\}^d$.

**Proof:** As in the proof of Lemma A.3, we assume that $d = 2d'$ and let $I = \{1, \cdots, d'\}, J = \{d'+1, \cdots, d\}$. Now we consider $\widetilde{I} = \{1, \cdots, d'R\}$ and a projection $\pi : \widetilde{I} \to I$ defined by $\pi(k) = \lfloor (k-1)/R \rfloor + 1$. Since we assumed that each elements of $\mathbf{u}$ and $\mathbf{v}$ can be re-sampled up to $R-1$ times, we can replace the probability Eq. (8) in the proof of Lemma A.3 by

$$\mathbb{P}\left( \not\exists i_1, i_2 \in \widetilde{I} \text{ s.t. } \pi(i_1) = \pi(i_2), \ |\widetilde{u}_{i_1} - 1| \leq \frac{\epsilon}{2}, \ |\widetilde{v}_{i_2} - w| \leq \frac{\epsilon}{2} \right), \tag{11}$$

where $\widetilde{u}_1, \cdots, \widetilde{u}_{d'R}, \widetilde{v}_1, \cdots, \widetilde{v}_{d'R} \sim U[-1,1]$. Since we have

$$\#\{(i_1, i_2) \in \widetilde{I} \times \widetilde{I} : \pi(i_1) = \pi(i_2)\} = d'R^2, \tag{12}$$

we can evaluate the probability Eq. (11) as

$$\text{Eq. (11)} \leq \left( 1 - \frac{\epsilon^2}{16} \right)^{d'R^2} \leq \frac{\delta}{2},$$

for $d' \geq \frac{16}{\epsilon^2 R^2} \log\left(\frac{\delta}{2}\right)$. The rest of the proof is same as Lemma A.3. $\qquad\square$

Then, we generalize the above lemma to the case which the target function $f(x)$ is a single-variable linear map with higher output dimensions.

**Lemma A.5** *Fix* $\epsilon, \delta \in (0,1), d_1, d_2 \in \mathbb{N}, \mathbf{w} \in [-1,1]^{d_2}$. *Let* $\mathbf{u} \sim U[-1,1]^{d_1}, V \sim U[-1,1]^{d_2 \times d_1}$ *be uniformly random weights of a 2-layered neural network* $g(x) := V\sigma(\mathbf{u} \cdot x)$. *Let* $R \in \mathbb{N}$ *and we assume that each elements of* $\mathbf{u}$ *and* $V$ *can be re-sampled with replacement up to* $R-1$ *times.*
*If* $d \geq 2\lceil \frac{16d_2}{\epsilon^2 R^2} \log(\frac{2d_2}{\delta}) \rceil$ *holds, then with probability at least* $1 - \delta$,

$$\|\mathbf{w} \cdot x - g_M(x)\|_2 \leq \epsilon, \text{ for all } x \in \mathbb{R}, |x| \leq 1, \tag{13}$$

*where* $g_M(x) := (V \odot M)^T \sigma(\mathbf{u} \cdot x)$ *for some* $M \in \{0,1\}^d$.

**Proof:** We denote $V = (V_{ki})_{1 \leq k \leq d_2, 1 \leq i \leq d_1}$. As in the proof of Lemma A.3, we assume $d_1 = 2d'_1$ and split the index set $\{1, \cdots, d_1\}$ into $I = \{1, \cdots, d'_1\}, J = \{d'_1 + 1, \cdots, d_1\}$. Also we consider the corresponding subnetworks of $g(x)$:

$$g_I(x) := \left( \sum_{i \in I} V_{ki}\sigma(u_i x) \right)_{1 \leq k \leq d_2}, \quad g_J(x) := \left( \sum_{i \in J} V_{ki}\sigma(u_i x) \right)_{1 \leq k \leq d_2}.$$

Similar as the proof of Lemma A.3 and Lemma A.4, it is enough to show that there probably exists a subnetwork of $g_I(x)$ which approximates $\mathbf{w} \cdot \sigma(x)$, and also that there simultaneously exists a subnetwork of $g_J(x)$ which approximates $-\mathbf{w} \cdot \sigma(-x)$.

For simplicity, we focus on $g_I(x)$ in the following argument, but same conclusion holds for $g_J(x)$ as well. Fix $k \in \{1, \cdots, d_2\}$. Then we consider the following probability,

$$\mathbb{P}\left(\not\exists i_1, i_2 \in \widetilde{I} \text{ s.t. } \pi(i_1) = \pi(i_2), \ |\widetilde{u}_{i_1} - 1| \leq \frac{\epsilon}{2\sqrt{d_2}}, \ |\widetilde{V}_{k,i_2} - w| \leq \frac{\epsilon}{2\sqrt{d_2}}\right), \tag{14}$$

where $\widetilde{u}_i, \widetilde{V}_{ki} \sim U[-1,1]$ for $i = 1, \cdots, d'R$. By using Eq. (12), if $d' \geq \frac{16d_2}{\epsilon^2 R^2} \log\left(\frac{2d_2}{\delta}\right)$, we have

$$\text{Eq. (14)} \leq \left(1 - \frac{\epsilon^2}{16d_2}\right)^{d'R^2} \leq \frac{\delta}{2d_2}$$

Therefore, by the union bound over $k = 1, \cdots, d_2$, we have $i_1, i_2 \in \widetilde{I}$ for each $k$ such that $i := \pi(i_1) = \pi(i_2)$, $|\widetilde{u}_{i_1} - 1| \leq \frac{\epsilon}{2\sqrt{d_2}}$, $|\widetilde{V}_{k,i_2} - w_k| \leq \frac{\epsilon}{2\sqrt{d_2}}$, with probability at least $1 - \delta$, and thus

$$\left|w_k \sigma(x) - V_{ki}\sigma(u_i x)\right| \leq \frac{\epsilon}{2\sqrt{d_2}}, \quad \text{for } x \in \mathbb{R}, |x| \leq 1, \tag{15}$$

if we substitute $\widetilde{u}_{i_1}$ for $u_i$, and $\widetilde{V}_{k,i_2}$ for $V_{ki}$. We note that the choice of $\widetilde{u}_{i_1}$ and $\widetilde{V}_{k,i_2}$ may not be unique, but Eq. (15) does not depend on these choice.

Therefore, by taking $M$ appropriately, we have

$$\begin{aligned} \|\mathbf{w}x - g_M(x)\|_2 &\leq \ \|\mathbf{w}\sigma(x) - g_I(x)\|_2 + \|-\mathbf{w}\sigma(-x) - g_J(x)\|_2 \\ &\leq \ \frac{\epsilon}{2} + \frac{\epsilon}{2} = \epsilon. \end{aligned}$$

for all $x \in \mathbb{R}$ with $|x| \leq 1$. $\qquad\qquad\qquad\qquad\qquad\qquad\qquad\qquad\qquad\qquad\qquad\quad$ $\square$

Subsequently, we can generalize Lemma A.5 to multiple variables version:

**Lemma A.6** *Fix* $\epsilon, \delta \in (0,1)$, $d_0, d_1, d_2 \in \mathbb{N}$, $W \in [-1,1]^{d_2 \times d_0}$. *Let* $U \sim U[-1,1]^{d_1 \times d_0}, V \sim U[-1,1]^{d_2 \times d_1}$ *be uniformly random weights of a 2-layered neural network* $g(\mathbf{x}) := V\sigma(U\mathbf{x})$. *Let* $R \in \mathbb{N}$ *and we assume that each elements of* $U$ *and* $V$ *can be re-sampled with replacement up to* $R - 1$ *times.*

*If* $d_1 \geq 2d_0 \lceil \frac{16d_0^2 d_2}{\epsilon^2 R^2} \log(\frac{2d_0 d_2}{\delta}) \rceil$ *holds, then with probability at least* $1 - \delta$,

$$\|W\mathbf{x} - g_{M,N}(\mathbf{x})\|_2 \leq \epsilon, \text{ for all } \mathbf{x} \in \mathbb{R}^{d_0}, \|x\|_\infty \leq 1, \tag{16}$$

*where* $g_{M,N}(x) := (V \odot M)^T \sigma((U \odot N) \cdot x)$ *for some* $M \in \{0,1\}^{d_2 \times d_1}$, $N \in \{0,1\}^{d_1 \times d_0}$.

**Proof:** Let $d'_1 = d_1/d_0$, and we assume $d'_1 \in \mathbb{N}$. We take $N$ as the following binary matrix:

$$N = \begin{pmatrix} \mathbf{1} & & 0 \\ & \ddots & \\ 0 & & \mathbf{1} \end{pmatrix}, \text{ where } \mathbf{1} = \begin{pmatrix} 1 \\ \vdots \\ 1 \end{pmatrix} \in \mathbb{R}^{d'_1 \times 1}$$

By the decomposition $U \odot N = \mathbf{u}_1 \oplus \cdots \oplus \mathbf{u}_{d_0}$, where each $\mathbf{u}_i$ is a $d'_1 \times 1$-matrix, we have

$$g_{M,N}(\mathbf{x}) = (V \odot M)^T \big(\sigma(\mathbf{u}_1 x_1) \oplus \cdots \oplus \sigma(\mathbf{u}_{d_0} x_{d_0})\big). \tag{17}$$

Here, we denote $M$ as follows:

$$M = \begin{pmatrix} M_1 & \cdots & M_{d_0} \end{pmatrix},$$

where each $M_i$ is a $d_2 \times d'_1$-matrix with binary coefficients. Then we have

$$V \odot M = (V_1 \odot M_1) + \cdots + (V_{d_0} \odot M_{d_0}), \quad V_i \in \mathbb{R}^{d_2 \times d'_1}. \tag{18}$$

By combining Eq. (17) and Eq. (18), we have

$$g_{M,N}(\mathbf{x}) = \sum_{1 \leq i \leq d_0} (V_i \odot M_i)\sigma(\mathbf{u}_i x_i). \tag{19}$$

Applying Lemma A.5 to each independent summands in Eq. (19), with probability at least $1 - \frac{\delta}{d_0}$, there exists $M_i$ for fixed $i \in \{1, \cdots, d_0\}$ such that

$$\|\mathbf{w}_i x_i - (V_i \odot M_i)\sigma(\mathbf{u}_i x_i)\|_2 \leq \frac{\epsilon}{d_0}, \text{ for } x_i \in \mathbb{R}, |x_i| \leq 1, \tag{20}$$

where $\mathbf{w}_i$ is the $i$-th column vector of $W$.

Using the union bound, we have $M_1, \cdots, M_{d_0}$ satisfying Eq. (20) simultaneously with probability at least $1 - \delta$. Therefore, by combining Eq. (19), Eq. (20) and the decomposition $W\mathbf{x} = \sum_j \mathbf{w}_j x_j$, we obtain Eq. (16). $\square$

Finally, by using Lemma A.6, we prove Theorem A.1. The outline of the proof is same as prior works [6][7].

**Proof of Theorem A.1:** By Lemma A.6, for each fixed $k \in \{1, \cdots, l\}$, we know that there exists binary matrices $M_{2k-1}, M_{2k}$ such that

$$\|F_k\mathbf{x} - (G_{2k} \odot M_{2k})\sigma\big((G_{2k-1} \odot M_{2k-1})\mathbf{x}\big)\|_2 \leq \frac{\epsilon}{2l}, \tag{21}$$

for all $\mathbf{x} \in \mathbb{R}^{d_0}$ with $\|\mathbf{x}\|_\infty \leq 1$, with probability at least $1 - \frac{\delta}{l}$. Taking the union bound, we can get $M = (M_1, \cdots, M_{2l})$ satisfying Eq. (21) for all $k = 1, \cdots, l$ with probability at least $1 - \delta$.

For the above $M$ and any $\mathbf{x}_0 \in \mathbb{R}^{d_0}$ with $\|\mathbf{x}_0\|_\infty \leq 1$, we define sequences $\{\mathbf{x}_k\}_{0 \leq k \leq l}$ and $\{\widetilde{\mathbf{x}}_k\}_{0 \leq k \leq l}$ as

$$\mathbf{x}_k := f_k(\mathbf{x}_{k-1}),$$
$$\widetilde{\mathbf{x}}_0 := \mathbf{x}_0, \quad \widetilde{\mathbf{x}}_k := g_{M,k}(\widetilde{\mathbf{x}}_{k-1}),$$

where $f_k(\mathbf{x})$ and $g_{M,k}(\mathbf{x})$ are given by

$$f_k(\mathbf{x}) := \begin{cases} \sigma\big(F_k\mathbf{x}\big), & (1 \leq k \leq l-1) \\ F_l\mathbf{x}, & (k = l) \end{cases}$$

$$g_{M,k}(\mathbf{x}) := \begin{cases} \sigma\big((G_{2k} \odot M_{2k})\sigma\big((G_{2k-1} \odot M_{2k-1})\mathbf{x}\big)\big), & (1 \leq k \leq l-1) \\ (G_{2l} \odot M_{2l})\sigma\big((G_{2l-1} \odot M_{2l-1})\mathbf{x}\big). & (k = l) \end{cases}$$

By induction on $k \in \{0, \cdots, l\}$, we can show that

$$\|\mathbf{x}_k - \widetilde{\mathbf{x}}_k\|_2 \leq \frac{k\epsilon}{l}, \quad \|\widetilde{\mathbf{x}}_k\|_\infty \leq 2. \tag{22}$$

For $k = 1$, this is trivial by definition. Consider the $k > 1$ case. First of all, we remark that the following inequality is obtained by Eq. (21) and the 1-Lipschitz property of ReLU function $\sigma$:

$$\|f_k(\mathbf{x}) - g_{M,k}(\mathbf{x})\|_2 \leq \frac{\epsilon}{2l}, \text{ for } \mathbf{x} \in \mathbb{R}^{d_0}, \|\mathbf{x}\|_\infty \leq 1.$$

By scaling $\mathbf{x}$, the above inequality can be rewritten as follows:

$$\|f_k(\mathbf{x}) - g_{M,k}(\mathbf{x})\|_2 \leq \frac{\epsilon}{l}, \text{ for } \mathbf{x} \in \mathbb{R}^{d_0}, \|\mathbf{x}\|_\infty \leq 2.$$

Then, we have

$$\begin{aligned}
\|\mathbf{x}_k - \widetilde{\mathbf{x}}_k\|_2 &= \|f_k(\mathbf{x}_{k-1}) - g_{M,k}(\widetilde{\mathbf{x}}_{k-1})\|_2 \\
&\leq \|f_k(\mathbf{x}_{k-1}) - f_k(\widetilde{\mathbf{x}}_{k-1}) + f_k(\widetilde{\mathbf{x}}_{k-1}) - g_{M,k}(\widetilde{\mathbf{x}}_{k-1})\|_2 \\
&\leq \|f_k(\mathbf{x}_{k-1}) - f_k(\widetilde{\mathbf{x}}_{k-1})\|_2 + \|f_k(\widetilde{\mathbf{x}}_{k-1}) - g_{M,k}(\widetilde{\mathbf{x}}_{k-1})\|_2 \\
&\leq \|F_k\|_{\text{Frob}} \cdot \|\mathbf{x}_{k-1} - \widetilde{\mathbf{x}}_{k-1}\|_2 + \frac{\epsilon}{l} \\
&\leq \frac{k\epsilon}{l} \quad \text{(by the induction hypothesis)},
\end{aligned}$$

and $\|\widetilde{\mathbf{x}}_k\|_\infty \leq \|\widetilde{\mathbf{x}}_k\|_2 \leq \|\mathbf{x}_k\|_2 + \|\mathbf{x}_k - \widetilde{\mathbf{x}}_k\|_2 \leq 1 + \frac{k\epsilon}{l} \leq 2$.

In particular, for $k = l$, Eq. (22) is nothing but Eq. (4). $\square$

# B    Details for our experiments

## B.1    Network architectures

In our experiments on CIFAR-10 and ImageNet, we used the following network architectures: Conv6, ResNet18, ResNet34, ResNet50, and ResNet101. In Table 1 (for CIFAR-10) and Table 2 (for ImageNet), we describe their configurations with a width factor $\rho \in \mathbb{R}_{>0}$. When $\rho = 1.0$, the architectures are standard ones.

For each ResNet network, the bracket $[\cdots]$ represents the basic block for ResNet18 and ResNet34, and the bottleneck block for ResNet50 and ResNet101, following the original settings by He et al. [3]. We have a batch normalization layer right after each convolution operation as well. Note that, when we train and evaluate these networks with IteRand or edge-popup [8], we replace the batch normalization to the non-affine one, which fixes its all learnable multipliers to 1 and all learnable bias terms to 0 following the design by Ramanujan et al. [8].

Table 1: Network Architectures for CIFAR-10. ($\rho$: a width factor)

| Layer \ Network | Conv6 | ResNet18 | ResNet34 |
|---|---|---|---|
| $3 \times 3$ Convolution & Pooling Layers | $64\rho$, $64\rho$, max-pool
$128\rho$, $128\rho$, max-pool
$256\rho$, $256\rho$, max-pool | 64, $[64\rho, 64\rho] \times 2$
$[128\rho, 128\rho] \times 2$
$[256\rho, 256\rho] \times 2$
$[512\rho, 512\rho] \times 2$
avg-pool | 64, $[64\rho, 64\rho] \times 3$
$[128\rho, 128\rho] \times 4$
$[256\rho, 256\rho] \times 6$
$[512\rho, 512\rho] \times 3$
avg-pool |
| Linear Layers | $256\rho$, $256\rho$, 10 | 10 | 10 |

Table 2: Network Architectures for ImageNet. ($\rho$: a width factor)

| Layer \ Network | ResNet18 | ResNet34 | ResNet50 | ResNet101 |
|---|---|---|---|---|
| Convolution | 64 ($7 \times 7$, stride 2) | | | |
| Pooling | max-pool ($3 \times 3$, stride 2, padding 1) | | | |
| Convolution Blocks | $[64\rho, 64\rho] \times 2$
$[128\rho, 128\rho] \times 2$
$[256\rho, 256\rho] \times 2$
$[512\rho, 512\rho] \times 2$ | $[64\rho, 64\rho] \times 3$
$[128\rho, 128\rho] \times 4$
$[256\rho, 256\rho] \times 6$
$[512\rho, 512\rho] \times 3$ | $[64\rho, 64\rho, 256\rho] \times 3$
$[128\rho, 128\rho, 512\rho] \times 4$
$[256\rho, 256\rho, 1024\rho] \times 6$
$[512\rho, 512\rho, 2048\rho] \times 3$ | $[64\rho, 64\rho, 256\rho] \times 3$
$[128\rho, 128\rho, 512\rho] \times 4$
$[256\rho, 256\rho, 1024\rho] \times 23$
$[512\rho, 512\rho, 2048\rho] \times 3$ |
| Pooling | avg-pool ($7 \times 7$) | | | |
| Linear | 1000 | | | |

## B.2    Hyperparameters

In our experiments, we trained several neural networks by three methods (SGD, edge-popup [8], and IteRand) on two datasets (CIFAR-10 and ImageNet). For each dataset, we adopted different hyperparameters as follows.

**CIFAR-10 experiments.**

- **SGD:** We used SGD with momentum for the optimization. It has the following hyperparameters: a total epoch number $E$, batch size $B$, learning rate $\eta$, weight decay $\lambda$, momentum coefficient $\mu$. For all network architectures, we used common values except for the learning rate and weight decay: $E = 100$, $B = 128$, and $\mu = 0.9$. For the learning rate and weight decay, we used $\eta = 0.01, \lambda = 1.0 \times 10^{-4}$ for Conv6, $\eta = 0.1, \lambda = 5.0 \times 10^{-4}$ for ResNet18 and ResNet34, following Ramanujan et al. [8]. Moreover, we decayed the learning rate by cosine annealing [4].

- **edge-popup:** With the same notation in Section 2, edge-popup has the same hyperparameters as SGD and an additional one, a sparsity rate $p$. We used the same values as SGD for each network except for the learning rate and the sparsity rate. For the learning rate, we used $\eta = 0.2$ for Conv6, and $\eta = 0.1$ for ResNet18 and ResNet34. We decayed the learning rate by cosine annealing, same as SGD. For the sparsity rate, we used $p = 0.5$ for Conv6, and $p = 0.6$ for ResNet18 and ResNet34.

- **IteRand:** With the notation in Section 3, IteRand has the same hypeparameters as edge-popup and the following additional ones: a randomization period $K_{\mathrm{per}} \in \mathbb{N}_{\geq 1}$ and a sampling rate $r \in [0, 1]$ for partial randomization. We used the same values as edge-popup for the former hyperparameters, and $K_{\mathrm{per}} = 300, r = 0.1$ for the latter ones.

**ImageNet experiments.**

- **SGD:** For all network architectures, we used the following hyperparameters (except for the learning rate): $E = 105, B = 128, \lambda = 1.0 \times 10^{-4}$, and $\mu = 0.9$. For the first 5 epochs, we gradually increased the learning rate as $\eta = 0.1 \times (i/5)$ for each $i$-th epoch ($i = 1, \cdots, 5$). For the next 95 epochs, we decayed the learning rate by cosine annealing starting from $\eta = 0.1$. For the final 5 epochs, we set the learning rate $\eta = 1.0 \times 10^{-5}$ to ensure the optimization converges.

- **edge-popup:** For all network architectures, we used the same hyperparameters as SGD and the sparsity rate $p = 0.7$.

- **IteRand:** For all network architectures, we used the same hyperparameters as edge-popup and $K_{\mathrm{per}} = 1000, r = 0.1$.

## B.3    Experimental results in table forms

**Figure 2 in Section 5:** In Table 3, we give means $\pm$ one standard deviations which are plotted in Figure 2 in Section 5.

Table 3: Results for Figure 2 in Section 5

| Network | Method | $\mathcal{D}_{\mathrm{param}}$ | $\rho = 0.25$ | $\rho = 0.5$ | $\rho = 1.0$ | $\rho = 2.0$ |
|---|---|---|---|---|---|---|
| Conv6 | IteRand | SC | $75.16 \pm 0.23$ | $84.31 \pm 0.13$ | $88.80 \pm 0.20$ | $90.89 \pm 0.17$ |
| | | KU | $73.90 \pm 0.47$ | $83.18 \pm 0.38$ | $88.20 \pm 0.38$ | $90.53 \pm 0.28$ |
| | edge-popup | SC | $70.35 \pm 1.16$ | $81.54 \pm 0.11$ | $87.60 \pm 0.11$ | $90.25 \pm 0.06$ |
| | | KU | $67.02 \pm 1.14$ | $79.37 \pm 0.23$ | $86.14 \pm 0.37$ | $89.91 \pm 0.23$ |
| | SGD | KU | $79.51 \pm 0.88$ | $84.46 \pm 0.34$ | $87.69 \pm 0.36$ | $89.63 \pm 0.22$ |
| ResNet18 | IteRand | SC | $86.09 \pm 0.24$ | $90.50 \pm 0.36$ | $92.61 \pm 0.17$ | $93.82 \pm 0.15$ |
| | | KU | $84.71 \pm 0.42$ | $89.96 \pm 0.06$ | $92.47 \pm 0.16$ | $93.52 \pm 0.16$ |
| | edge-popup | SC | $84.23 \pm 0.33$ | $89.61 \pm 0.06$ | $92.29 \pm 0.04$ | $93.57 \pm 0.13$ |
| | | KU | $81.13 \pm 0.09$ | $88.45 \pm 0.53$ | $91.79 \pm 0.19$ | $93.25 \pm 0.06$ |
| | SGD | KU | $90.99 \pm 0.29$ | $93.10 \pm 0.14$ | $94.43 \pm 0.05$ | $95.02 \pm 0.23$ |
| ResNet34 | IteRand | SC | $88.26 \pm 0.35$ | $91.87 \pm 0.17$ | $93.54 \pm 0.27$ | $94.25 \pm 0.15$ |
| | | KU | $87.36 \pm 0.14$ | $91.58 \pm 0.16$ | $93.27 \pm 0.19$ | $94.05 \pm 0.10$ |
| | edge-popup | SC | $87.37 \pm 0.30$ | $91.41 \pm 0.26$ | $93.27 \pm 0.09$ | $94.25 \pm 0.13$ |
| | | KU | $85.31 \pm 0.66$ | $90.85 \pm 0.11$ | $93.00 \pm 0.11$ | $93.96 \pm 0.10$ |
| | SGD | KU | $92.26 \pm 0.12$ | $94.20 \pm 0.25$ | $94.66 \pm 0.18$ | $95.33 \pm 0.10$ |

**Figure 3 in Section 5:** In Table 4, we give means $\pm$ one standard deviations which are plotted in Figure 3 in Section 5.

Table 4: Results for Figure 3 in Section 5

| Network | Method | $\mathcal{D}_{\mathrm{param}}$ | Accuracy (Top-1) | Sparsity (%) | # of Params |
|---|---|---|---|---|---|
| ResNet18 | SGD | KU | 69.89 | 0% | 11.16M |
| ResNet34 | SGD | KU | 73.82 | 0% | 21.26M |
| ResNet34 | IteRand | SC | 65.86 | 70% | 6.38M |
| | edge-popup | SC | 62.14 | 70% | 6.38M |
| ResNet50 | IteRand | SC | 69.19 | 70% | 7.04M |
| | edge-popup | SC | 67.11 | 70% | 7.04M |
| ResNet101 | IteRand | SC | 72.99 | 70% | 12.72M |
| | edge-popup | SC | 71.85 | 70% | 12.72M |

# C  Additional experiments

## C.1  Ablation study on hyperparameters: $K_{\mathrm{per}}$ and $r$

IteRand has two hyperparameters: $K_{\mathrm{per}}$ (see line 5 in Algorithm 3) and $r$ (see Eq. (3) in Section 3).

$K_{\mathrm{per}}$ controls the frequency of randomizing operations during the optimization. Note that, in our experiments in Section 5, we fixed it to $K_{\mathrm{per}} = 300$ on CIFAR-10, which is nearly 1 epoch ($= 351$ iterations) when the batch size is 128. As we discussed in Section 3, too small $K_{\mathrm{per}}$ may degrade the performance because it may randomize even the important weights before their scores are well-optimized. In contrast, too large $K_{\mathrm{per}}$ makes IteRand almost same as edge-popup, and thus the effect of the randomization disappears. Figure 1 shows this phenomenon with varying $K_{\mathrm{per}}$ in $\{1, 30, 300, 3000, 30000\}$. In relation to our theoretical results (Theorem A.1), we note that the expected number (here we denote $R'$) of randomizing operations for each weight is in inverse proportion to $K_{\mathrm{per}}$. The theoretical results imply that greater $R'$ leads to better approximation ability of IteRand. We observe that the results in Figure 1 are consistent with this implication, in the region where $K_{\mathrm{per}}$ is not too small ($K_{\mathrm{per}} \geq 300$).

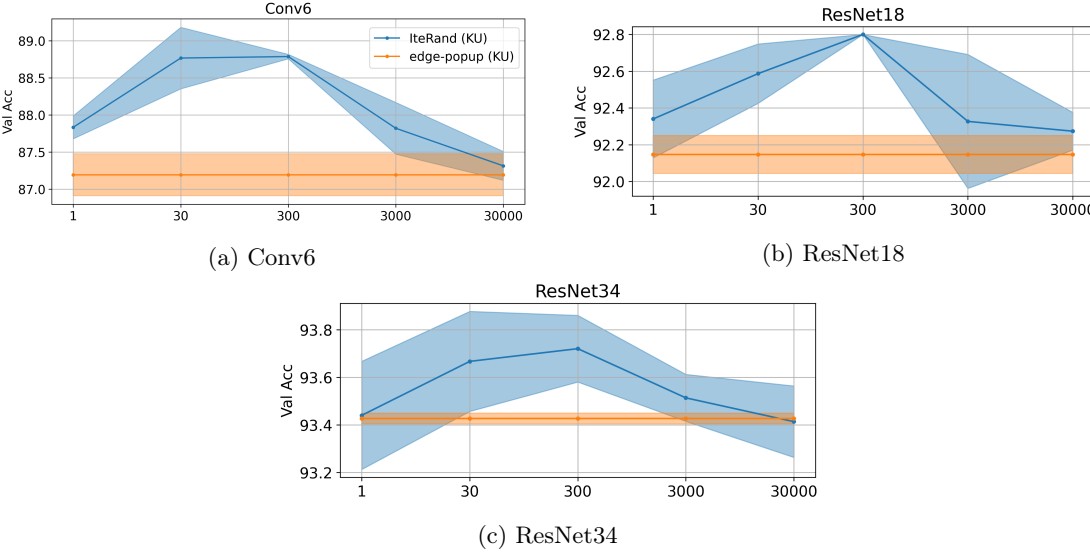

(a) Conv6

(b) ResNet18

(c) ResNet34

Figure 1: We train and validate Conv6, ResNet18 and ResNet34 on CIFAR-10 with various $K_{\mathrm{per}} \in \{1, 30, 300, 3000, 30000\}$. The x-axis is $K_{\mathrm{per}}$ in a log scale, and the y-axis is validation accuracy.

Also, we investigate the relationship between $K_{\mathrm{per}}$ and $r$. Figure 2 shows how test accuracy changes when both $K_{\mathrm{per}}$ and $r$ vary. From this result, we find that the accuracies seem to depend on $r/K_{\mathrm{per}}$. This may be because each pruned parameter in the neural network is randomized $Nr/K_{\mathrm{per}}$ times in expectation during the optimization. On the other hand, when we use larger $r \in [0, 1]$,

we have to explore $K_{\mathrm{per}}$ in longer period (e.g. 3000 iterations when $r = 1.0$). Thus appropriately choosing $r$ leads to shrink the search space of $K_{\mathrm{per}}$.

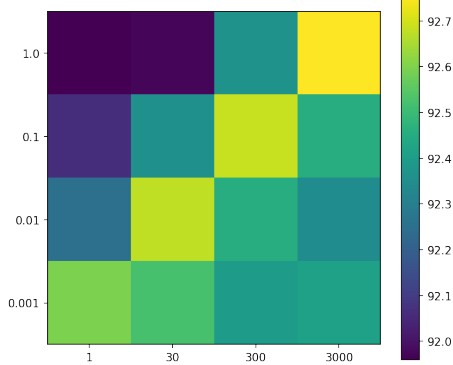

Figure 2: Test accuracies on CIFAR-10 with ResNet18. The x-axis is $K_{\mathrm{per}} \in \{1, 30, 300, 3000\}$ and the y-axis is $r \in \{0.001, 0.01, 0.1, 1.0\}$.

## C.2 Computational overhead of iterative randomization

IteRand introduces additional computational cost to the base method, edge-popup, by iterative randomization. However, the additional computational cost is negligibly small in all our experiments. We measured the average overhead of a single randomizing operation, which is the only difference from edge-popup, as follows: 97.10 ms for ResNet18 (11.2M params), 200.55 ms for ResNet50 (23.5M params). Thus, the total additional cost should be about 10 seconds in the whole training (1.5 hours) for ResNet18 on CIFAR-10 and 200–300 seconds in the whole training (one week) for ResNet50 on ImageNet.

Also, we measured the total training times of our expriments for ResNet-18 on CIFAR-10 (Table 5). The additional computational cost of IteRand over edge-popup is tens of seconds, which is quite consistent with the above estimates.

Table 5: Training times for ResNet18 on CIFAR-10.

| | |
|---|---|
| IteRand | 6253.69 (secs) |
| edge-popup | 6231.80 (secs) |
| SGD | 6388.61 (secs) |

## C.3 Experiments with varying sparsity

Table 6 shows the comparison of IteRand and edge-popup when varying the sparsity parameter $p \in \{0.1, 0.3, 0.5, 0.7, 0.9, 0.95, 0.99\}$. We can see that IteRand is effective for almost all the sparsities $p$.

Table 6: Test accuracies with various sparsities on CIFAR-10.

| Networks | Methods | $p = 0.1$ | $p = 0.3$ | $p = 0.5$ | $p = 0.7$ | $p = 0.9$ | $p = 0.95$ | $p = 0.99$ |
|---|---|---|---|---|---|---|---|---|
| Conv6 | IteRand (SC) | **87.17** ± 0.21 | **89.08** ± 0.14 | **89.19** ± 0.13 | **87.67** ± 0.07 | **76.09** ± 0.23 | **59.03** ± 1.55 | 12.04 ± 3.40 |
| | edge-popup (SC) | 80.31 ± 0.27 | 86.55 ± 0.22 | 87.57 ± 0.03 | 86.40 ± 0.13 | 73.25 ± 0.79 | 55.23 ± 1.48 | **12.13** ± 3.46 |
| ResNet18 | IteRand (SC) | **91.79** ± 0.20 | **92.67** ± 0.11 | **92.66** ± 0.22 | **92.61** ± 0.15 | **91.82** ± 0.15 | **90.40** ± 0.42 | **76.31** ± 2.56 |
| | edge-popup (SC) | 87.37 ± 0.18 | 91.43 ± 0.16 | 92.25 ± 0.18 | 92.32 ± 0.10 | 91.64 ± 0.18 | 90.28 ± 0.30 | 75.21 ± 2.71 |

Moreover, we compared the pruning-only approach (IteRand and edge-popup) and the iterative magnitude pruning (IMP) approach [2] with various sparsity rates. We employed the OpenLTH framework [1], which contains the implementation of IMP, as a codebase for this experiment and implemented both edge-popup and IteRand in this framework. The results are shown in Table 7.

Overall, the IMP outperforms the pruning-only methods. However, there is still room for improvement in the pruning-only approach such as introducing scheduled sparsities or an adaptive threshold, which is left to future work.

Table 7: Comparison of the pruning-only approach and magnituide-based one.

| Networks | Methods | $p = 0.5$ | $p = 0.7$ | $p = 0.9$ | $p = 0.95$ | $p = 0.99$ |
|---|---|---|---|---|---|---|
| | IteRand (SC) | $88.46 \pm 0.22$ | $88.29 \pm 0.42$ | $87.05 \pm 0.07$ | $84.37 \pm 0.59$ | $64.93 \pm 4.81$ |
| VGG11 | edge-popup (SC) | $87.09 \pm 0.31$ | $87.34 \pm 0.21$ | $85.11 \pm 0.55$ | $81.06 \pm 0.84$ | $61.71 \pm 6.05$ |
| | IMP with 3 retraining [2] | $\mathbf{91.47} \pm 0.15$ | $\mathbf{91.48} \pm 0.16$ | $\mathbf{90.89} \pm 0.08$ | $\mathbf{90.39} \pm 0.27$ | $\mathbf{88.076} \pm 0.17$ |
| | IteRand (SC) | $84.17 \pm 0.78$ | $82.31 \pm 0.52$ | $70.96 \pm 0.55$ | $55.60 \pm 0.70$ | $24.45 \pm 0.67$ |
| ResNet20 | edge-popup (SC) | $76.57 \pm 0.91$ | $75.83 \pm 2.75$ | $49.25 \pm 6.33$ | $42.96 \pm 3.91$ | $20.11 \pm 3.43$ |
| | IMP with 3 retraining [2] | $\mathbf{90.70} \pm 0.37$ | $\mathbf{89.79} \pm 0.14$ | $\mathbf{86.87} \pm 0.26$ | $\mathbf{84.28} \pm 0.08$ | $\mathbf{71.78} \pm 1.66$ |

## C.4 Detailed empirical analysis on the parameter efficiency

We conducted experiments to see how much more network width edge-popup requires than IteRand to achieve the same accuracy (Table 8). Here we use ResNet18 with various width factors $\rho \in \mathbb{R}_{>0}$. We first computed the test accuracies of IteRand with $\rho = 0.5, 1.0$ as target values. Next we explored the width factors for which edge-popup achieves the same accuracy as the target values. Table 8 shows that edge-popup requires 1.3 times wider networks than IteRand in this specific setting.

Table 8: Test accuracies for ResNet18 with various width factors.

| | $\rho = 0.5$ | $\rho = 0.65$ | $\rho = 1.0$ | $\rho = 1.3$ |
|---|---|---|---|---|
| IteRand (KU) | $\mathbf{89.96} \pm 0.06$ | - | $\mathbf{92.47} \pm 0.16$ | - |
| edge-popup (KU) | $88.45 \pm 0.53$ | $89.99 \pm 0.08$ | $91.79 \pm 0.19$ | $\mathbf{92.54} \pm 0.19$ |

## C.5 Experiments with large-scale networks

In addition to the experiments in Section 5, we conducted experiments to see the effectiveness of IteRand with large-scale networks: WideResNet-50-2 [10] and ResNet50 with the width factor $\rho = 2.0$. Table 9 shows that the iterative randomization is still effective for these networks to improve the performance of weight-pruning optimization.

Table 9: Experiments with large-scale networks.

| | IteRand (SC) | edge-popup (SC) | # of parameters |
|---|---|---|---|
| WideResNet-50-2 | $\mathbf{73.57}\%$ | $71.59\%$ | 68.8 M |
| ResNet50 ($\rho = 2.0$) | $\mathbf{74.05}\%$ | $72.96\%$ | 97.8 M |

## C.6 Experiments with a text classification task

Although our main theorem (Theorem 4.1) indicates that the effectiveness of IteRand does not depend on any specific tasks, we only presented the results on image classification datasets in the body of this paper. In Table 10, we present experimental results on a text classification dataset, IMDB [5], with recurrent neural networks (see Table 11 for the network architectures). For this experiment, we implemented both edge-popup and IteRand on the Jupyter notebook originally written by Trevett [9]. All models are trained for 15 epochs and the learning rate $\eta$ we used is $\eta = 1.0$ for SGD and $\eta = 2.5$ for edge-popup and IteRand. Note that the learning rate $\eta = 2.5$ does not work well for SGD, thus we employed the different value from the one for edge-popup and IteRand. Also we set the hyperparameters for IteRand as $p = 0.5$, $K_{\mathrm{per}} = \lfloor 270/6 \rfloor$ ($\approx 1/6$ epochs) and $r = 1.0$.

Table 10: Test accuracies on the IMDB dataset over 5 runs.

| Networks     Methods | IteRand (SC) | edge-popup (SC) | SGD |
|---|---|---|---|
| LSTM | $\mathbf{88.44} \pm 0.28\%$ | $88.16 \pm 0.12\%$ | $87.39 \pm 0.37\%$ |
| BiLSTM | $\mathbf{88.51} \pm 0.24\%$ | $88.34 \pm 0.19\%$ | $87.62 \pm 0.22\%$ |

Table 11: The network architectures for IMDB.

| Layer     Network | (Bi)LSTM |
|---|---|
| Embedding Layer | $\text{dim} = 100$ |
| LSTM Layer | $\text{hidden\_dim} = 256, \text{num\_layers} = 1,$ (bidirectional = True for BiLSTM) |
| Dropout Layer | $p = 0.2$ |
| Linear Layer | $\text{output\_dim} = 1$ |
| Output Layer | sigmoid |