# OpenReview forum: "Pruning Randomly Initialized Neural Networks with Iterative Randomization"
_NeurIPS.cc/2021/Conference — NeurIPS 2021 Spotlight_

### Official Review · Reviewer_5mau · 2021-07-03

**Rating:** 7
**Confidence:** 3

**Summary:**

This work proposes an improved method to find a performative subnetwork within a randomly weighted neural network. In contrast to prior work, a fraction of unused weights are periodically reset. This is a novel idea with empirical improvements and theoretical justification.

**Limitations And Societal Impact:**

Authors adequately addressed the limitations and potential negative societal impact.

**Main Review:**

Overall the paper is clear to follow and enjoyable to read. The method is interesting, well motivated, and there are empirical gains. A few suggestions which could improve quality:

1) Additional empirical results would strengthen the paper's message. For instance, this work is introducing two new hyperparameters (r and K_per), though there is only an ablation on one of these parameters. How did the authors choose K_per? Moreover, edge-popup could be added to Figure 3 -- it does seem as though this work performs better.

2) There are a couple points where clarity in limitations could be improved. For instance, is IteRand just has an SC initialization but is no longer SC by the end of training. This is a shortcoming as weights can no longer be encoded with in binary. Moreover, with edge-popup or Zhou et al., a single random seed can be stored which generates all the weights, it seems this is no longer the case with IteRand.

Minor comments:
- Line 32-33 I believe Pensia et al. show that their theoretical result holds with log overaparam and not that this is required emperically.

-- Added --

I acknowledge the response and have raised my score to 7

**Time Spent Reviewing:**

2.5h

---

> ### Author Response · Authors · 2021-08-10
> **Replies for your valuable review**
>
>
> Thanks for your interest in our paper and helpful suggestions.
>
> > 1. Additional empirical results would strengthen the paper's message. For instance, this work is introducing two new hyperparameters (r and K_per), though there is only an ablation on one of these parameters. How did the authors choose K_per?
>
> In addition to an ablation for the hyperparameter $r$ (Figure 1), we also performed one for $K_{\rm per}$ (Figure 1 in appendix.pdf, Supplementary Material). This indicates that $K_{\rm per}$ should be neither too small nor too large. If $K_{\rm per}$ is too small, then the optimization may be unstable and also the computational cost increases. On the contrary, if $K_{\rm per}$ is too large, the number of re-sampling will be insufficient. To choose the hyperparameters, we examined several pairs $(r, K_{\rm per})$ and found out that $(r,K_{\rm per})=(0.1,300)$ works well on CIFAR-10 and $(r,K_{\rm per})=(0.1,1000)$ on ImageNet, in terms of both accuracy and computational cost. Even so, additional ablation that both $r$ and $K_{\rm per}$ are simultaneously varied would certainly be helpful. We will add such ablation analyses in the final version.
>
>
> > Moreover, edge-popup could be added to Figure 3 -- it does seem as though this work performs better.
>
> We assume that the reviewer is talking about Figure 1 instead of Figure 3, since Figure 3 already contains the results for edge-popup. In Figure 1, we performed an ablation analysis on the hyperparameter $r\in \\{0.0, 0.01, 0.1, 1.0\\}$ of IteRand. Although edge-popup does not seem to appear in Figure 1, the case of $r=0.0$ is actually edge-popup. (Also see lines 129-131.)
>
> If we have misunderstood your suggestion, it would be helpful if you could give us more details.
>
> > 2. There are a couple points where clarity in limitations could be improved. For instance, is IteRand just has an SC initialization but is no longer SC by the end of training. This is a shortcoming as weights can no longer be encoded with in binary.
>
> This is not the case. We understand that the reviewer concerns that the weights obtained by IteRand do not maintain its initial binarity when they are initialized and randomized with an SC distribution, which is a uniform distribution over a binary set $\\{-C, C\\}$ for some $C\in\mathbb{R}$. However, in general, when we use a distribution $\mathcal{D}$ over any fixed subset $S \subset \mathbb{R}$, the randomizing operation (Eq. 3) always returns a value in $S$. Therefore, **IteRand maintain the binarity of the weights when using an SC distribution**.
>
> Also if we have misunderstood your concern, it would be helpful if you could give us more details.
>
> > Moreover, with edge-popup or Zhou et al., a single random seed can be stored which generates all the weights, it seems this is no longer the case with IteRand.
>
> Thanks for bringing up the interesting point. We agree with this comment. A model trained by edge-popup can be stored efficiently as a pair of a single random seed and binary matrices, instead of a pair of weight matrices and binary matrices. In the same way, a model trained by IteRand can be stored as a single random seed and $R$ sets of binary matrices, where $R$ is the number of re-samplings. In our experiments, $R\approx 100$ for CIFAR-10 and $R\approx 1K$ for ImageNet. Although we can control $R$ by changing the hyperparameter $K_{\rm per}$, there may be a tradeoff between accuracy and storage efficiency. (Also see Figure 1 in appendix.pdf, Supplementary Material.) We would like to add this argument in Limitations section (Section 7).
>
> > Minor comments: Line 32-33 I believe Pensia et al. show that their theoretical result holds with log overaparam and not that this is required emperically.
>
> We agree with this comment. We will rewrite lines 32-33 to emphasize the theoretical aspect of their work, as follows:
> ```
> (before) Pensia et al. [22] showed that the required network width for the weight-pruning optimization needs to be logarithmically wider than the weight-optimization at least in the case of shallow networks.
> (after) Pensia et al. [22] theoretically showed that the required network width for the weight-pruning optimization needs to be logarithmically wider than the weight-optimization at least in the case of shallow networks.
> ```

---

> > ### Comment · Reviewer_5mau · 2021-08-30
> > **Thank you.**
> >
> > Thank you very much, this addresses my concerns. I've raised my score to 7.
> >
> > Would be good to acknowledge the limitation with regards to storing the weights as a seed in the paper.

---

### Official Review · Reviewer_HPM9 · 2021-07-16

**Rating:** 7
**Confidence:** 3

**Summary:**

This paper proposes a memory-efficient approach for \textit{weight-pruning optimization}, which suffers from a much larger memory cost than standard \textit{weight optimization}.

The proposed approach, weight-pruning with iterative randomization (IteRand), decreases the required network width by introducing pruned weights randomization at each iteration. The author theoretically shows that, with sufficient randomizing operations, the width can be reduced to constant factors of that of a standard network trained by weight-optimization.

The empirical experiments demonstrate IteRand achieves better performance than existing weight-pruning optimization methods under the same memory budget.

**Ethical Concerns:**

There are no ethical issues with this paper.

**Limitations And Societal Impact:**

The authors adequately addressed the limitations and potential negative societal impact of their work.

**Main Review:**

**Reason to accept**
- The paper is well-organized and finely written.
- The limitations of the existing works are thoroughly discussed and the paper is clearly motivated.
- The authors both theoretically prove and empirically verify that IteRand can effectively reduce memory consumption and increase the complexity of the network.

**Reason to reject & Questions**
- Section 5.1&5.2: The gain of IteRand seems to decreases as the network size scales up (ResNet34 > ResNet50 > ResNet101 in Figure 3(a), and the same phenomenon in Figure 2.). I wonder how IteRand performs in settings where the memory constraint is an actual consideration (e.g., when training a model with hundreds of million parameters).
- How much is the additional computational cost of IteRand over edge-popup?
- I wonder if the authors can show some empirical results on 1) given the same size of $f(x)$, what is the width required for $g(x)$ for IteRand and for EdgePop to achieve the same performance? This may help understand the percentage of memory that is actually saved. 2) the performance under different $p$s to verify the consistency of the improvement.

Disclaimer: I am not an expert in theoretical proofs. All my evaluation assumes the correctness of Theorem 4.1 and is subject to change if the proof turns out to be erroneous.

**Time Spent Reviewing:**

2h

---

> ### Author Response · Authors · 2021-08-10
> **Replies for your valuable review**
>
>
> Thanks for your positive review and fruitful questions.
>
> > I wonder how IteRand performs in settings where the memory constraint is an actual consideration (e.g., when training a model with hundreds of million parameters).
>
> After the first submission, we conducted additional experiments on ImageNet with larger networks than reported in our paper: WRN-50-2 [1] and ResNet-50 with the width factor $\rho=2.0$. The latter network has nearly a hundred of million parameters. The results are the following:
>
> |                         | IteRand (SC) | edge-popup (SC) | # of params before pruning |
> | --------                | :--------:   | :-------: | :-------:  |
> | WRN-50-2 [1]            | ${\bf 73.57}\\%$ |  $71.59\\%$   | $68.8$ M |
> | ResNet-50 $(\rho=2.0)$  |  ${\bf 74.01} \\%$ |  $72.96 \\%$ |  $97.8$ M  |
>
> For both networks, our method still boosts test accuracies compared to the baseline method, edge-popup. The results suggest that IteRand works well under an actual memory constraint. Also we expect that more hyperparameter tuning ($r$ and $K_{\rm per}$) for each network can help to achieve more performance gain by IteRand. We plan to add these results to the final version of our paper.
>
> > How much is the additional computational cost of IteRand over edge-popup?
>
> To come to the point, the additional computational cost is **negligibly small** since the randomization operation can be performed efficiently by GPU. Note that the only difference between IteRand and edge-popup is the randomization operation (Line 6 in Algorithm 3). To estimate the overhead of the operation, we performed 1000 runs of the randomization operation alone and calculated the average computation time: **97.10 ms** for ResNet-18 (11.2M params) and **200.55 ms** for ResNet-50 (23.5M params). In all experiments in our paper, the operation was executed for $R$ times, where $R\approx 100$ on CIFAR-10 and $R\approx 1000$ on ImageNet, during the whole training. Therefore,  the additional computational cost of IteRand can be estimated as: **about 10 seconds in the whole training ($\approx$ 1.5 hours) for ResNet18 on CIFAR-10 and 200~300 seconds in the whole training ($\approx$ one week) for ResNet50 on ImageNet**. Also, as we answered to Reviewer **x4sx**, the actual training times for ResNet18 on CIFAR-10 are 6231.80 secs (edge-popup) and 6253.69 secs (IteRand). The difference is 21.89 secs and thus matches the above estimates.
>
> We would like to add the above discussion and results on the computational overhead to our paper.
>
> > I wonder if the authors can show some empirical results on 1) given the same size of $f(x)$, what is the width required for $g(x)$ for IteRand and for EdgePop to achieve the same performance? This may help understand the percentage of memory that is actually saved.
>
> To answer your question, we consider the following concrete problem: For ResNet-18 with varying width factor $\rho \in \mathbb{R}_{>0}$, when do IteRand and edge-popup achieve a fixed test accuracy on CIFAR-10? Figure 2 in our paper provides some insights into this problem. For simplicity, we compare `IteRand (KU)` and `edge-popup (KU)`. Looking at `IteRand (KU)` for ResNet-18 in Figure2, we can see $\rho=0.5$ achieves about $90\\%$ test accuracy and $\rho=1.0$ achieves about $92.5\\%$. For `edge-popup (KU)`, we conducted additional experiments to investigate when it achieves the same accuracies ($90\\%$ and $92.5\\%$). The results are the following:
>
> | ResNet18 w/ width factor $\rho$ | $\rho=0.5$ | $\rho=0.65$ | $\rho=1.0$ | $\rho=1.3$ |
> | --- | :---: | :---: | :---: | :---: |
> |IteRand (KU)|$\mathbf{89.96}\pm0.06$|--|$\mathbf{92.47}\pm0.16$|--|
> |edge-popup (KU)|$88.45\pm0.53$| $\mathbf{89.99}\pm0.08$ |$91.79\pm0.19$|$\mathbf{92.54}\pm 0.19$|
>
> From these results, for `edge-popup (KU)`, we can see $\rho=0.65$ achieves about $90\\%$ and $\rho=0.7$ achieves about $92.5\\%$. Therefore, **for ResNet-18 on CIFAR-10, edge-popup requires at least $1.3$ times wider networks than IteRand**. We would like to add this analysis to the final paper.
>
> >  2\) the performance under different $p$s to verify the consistency of the improvement.
>
> We conducted additional experiments with various pruning rates, for Conv6 and ResNet-18 on CIFAR-10. The results are in the following table. Note that a higher $p$ means the network after pruned is more sparse. Overall, **IteRand consistently improves accuracies with different pruning rates**. We will add these results to Appendix for the final paper.
>
> | Conv6 w/ sparsity $p$ | $p=0.1$ | $p=0.3$ | $p=0.5$ | $p=0.7$ | $p=0.9$ | $p=0.95$ | $p=0.99$ |
> | -------- |:----:|:----:|:----:|:----:|:----:|:----:|:----:|
> |IteRand (SC)|$\mathbf{87.17}\pm0.21$|$\mathbf{89.08}\pm0.14$|$\mathbf{89.19}\pm0.13$|$\mathbf{87.67}\pm0.07$|$\mathbf{76.09}\pm0.23$|$\mathbf{59.03}\pm1.55$|$12.04\pm3.40$ (failed)|
> |edge-popup (SC)|$80.31\pm0.27$|$86.55\pm0.22$|$87.57\pm0.03$|$86.40\pm0.13$|$73.25\pm0.79$|$55.23\pm1.48$|$\mathbf{12.13}\pm3.46$ (failed)|
>
> | ResNet18 w/ sparsity $p$ |$p=0.1$ | $p=0.3$ | $p=0.5$ | $p=0.7$ | $p=0.9$ | $p=0.95$ | $p=0.99$ |
> | -------- | :--------: | :--------: | :--------: | :--------: |:--------: |:--------: |:--------: |
> |IteRand (SC)|$\mathbf{91.76}\pm0.11$|$\mathbf{92.69}\pm0.11$|$\mathbf{92.8}\pm0.07$|$\mathbf{92.61}\pm0.19$|$\mathbf{91.81}\pm 0.15$|$\mathbf{90.61}\pm0.44$|$\mathbf{76.91}\pm2.97$|
> |edge-popup (SC)|$87.41\pm0.21$|$91.49\pm0.08$|$92.37\pm0.14$|$92.29\pm0.10$|$91.75\pm0.10$|$90.29\pm0.41$|$73.69\pm2.21$|
>
>
> ## References
>
> [1] S. Zagoruyko and N. Komodakis. "Wide Residual Networks," BMVC 2016.

---

> > ### Comment · Reviewer_HPM9 · 2021-08-30
> > **Thanks for the additional experiments**
> >
> > I thank the authors for adding these detailed experiments. The provided results are interesting and clear my major concern on the effectiveness and time cost of applying this method on larger-scale networks and data.

---

### Official Review · Reviewer_x4sx · 2021-07-16

**Rating:** 7
**Confidence:** 4

**Summary:**

The paper proposes a new technique to reduce the network width required for pruning based algorithms. In particular, instead of starting from a wider network and then pruning weights to train the network, the paper proposes that pruned neurons can be resampled with random weights. Hence, instead of having extra weights in the form of extra width from the beginning, the new technique brings in new weights every time some of the existing weights are pruned. The motivation behind this is to reduce the memory footprint of the network before the pruning.

The idea and the empirical results are good, however, the theoretical result could be improved significantly.

**Limitations And Societal Impact:**

1. Please compare your results with techniques that use both SGD and pruning iteratively. Even though (SGD+pruning) might outperform purely pruning based techniques, like the one proposed in this paper, it is still a necessary evaluation that should be included in the paper.
2. Please provide the comparisons for the actual training time (in seconds) for different techniques.
3. Please provide details on the number of re-samplings that happened during the training.

**Main Review:**

The new idea that pruned weights can be resampled seems very intuitive and effective. The empirical results suggest that this indeed outperforms existing pruning techniques. However, the authors should provide more empirical evidence like actual training time, comparison with state-of-the-art (SGD+pruning) techniques like the lottery ticket hypothesis, and the actual number of times weights were resampled.

The theoretical result does not provide a strong support for the technique. In particular, even though the result is purely existential, the requirement on the number of re-samplings, $R$, is huge. It is proportional to the network depth and width. This seems to suggest that the procedure might not be efficient (in terms of computational time) for training large networks, since each resampling would need at least one complete iteration.

The paper is well written and easy to understand.

Overall, the proposed technique is novel and can improve the state-of-the-art in pruning techniques.

**Time Spent Reviewing:**

5

---

> ### Author Response · Authors · 2021-08-10
> **Replies for your valuable review**
>
> Thanks for your positive review and helpful comments.
>
> > 1. Please compare your results with techniques that use both SGD and pruning iteratively. Even though (SGD+pruning) might outperform purely pruning based techniques, like the one proposed in this paper, it is still a necessary evaluation that should be included in the paper.
>
> Thanks for the insightful suggestion. Following your suggestion, we are preparing the comparison between our pruning-only approach and the magnitude-based pruning approach. So far we have almost no experimental results yet, but we can predict the results based on the existing literature [1] and our current experimental results:
>
> - In the low sparsity regime (with pruning rates $0.0 < p < 0.1$), the magnitude-based approach keeps almost same accuracy as a densely trained model, whereas the pruning-only approach gets closer to a random model ($p=0.0$).
> - In the extremely high sparsity regime (with pruning rates $0.9 < p < 1.0$), both edge-popup and IteRand will be significantly inferior to the iterative magnitude pruning, because both of the previous work [1] and our paper does not focus mainly on the sparsity. Indeed there is still room for improvement such as introducing scheduled sparsities or an adaptive threshold, which are left to future work.
>
> We will add the comparison between these approaches to the final version of our paper.
>
> > 2. Please provide the comparisons for the actual training time (in seconds) for different techniques.
>
> We measured the execution times of our expriments for ResNet-18 on CIFAR-10 for example:
>
> - edge-popup: 6231.80 (secs)
> - IteRand: 6253.69 (secs)
> - SGD: 6388.61 (secs)
>
> The additional computational cost of IteRand over edge-popup is tens of seconds, which is quite consistent with our estimates in the response to Reviewer **HPM9**. Also, we observed that SGD was slightly slow compared to edge-popup and IteRand, which may be caused by the dense feed-forward computation and learnable batch normalization layers. We will add these analysis to the final version of our paper.
>
> > 3. Please provide details on the number of re-samplings that happened during the training.
>
> As we described in lines 106-107 (also see lines 5-7 in Algorithm 3), each re-sampling step is executed once per $1$ epoch for CIFAR-10 and once per $1/10$ epochs for ImageNet. Thus the total number of re-samplings is about $100$ times for CIFAR-10 and $1$K for ImageNet during each training. Also, note that the number of resamplings can be calculated by the following formula: $N / K_{\rm per}$, where $N$ is the number of total iterations during training and $K_{\rm per}$ is the hyperparameter of re-sampling frequency. We would like to add this description to the final version.
>
> > In particular, even though the result is purely existential, the requirement on the number of re-samplings, R, is huge. It is proportional to the network depth and width. This seems to suggest that the procedure might not be efficient (in terms of computational time) for training large networks, since each resampling would need at least one complete iteration.
>
> We agree with your concern about the sufficient condition on $R$ for the required width factor $2\lceil O(.../R^2)\rceil$ (in Theorem 4.1) to be the constant $2$. However, in fact, all our experiments are conducted with smaller $R$: $R\approx 100$ for CIFAR-10 and $R \approx 1K$ for ImageNet. As a result, as shown in the above execution times, the computational overhead caused by iterative randomization is **negligibly small** in the whole training time. Indeed, a single randomizing operation takes only hundreds of milliseconds even for ResNet-50. Therefore, the total additional costs by randomization is just hundreds of seconds (for ResNet-50 on ImageNet) compared to the whole training time ($\approx$ one week).
>
> Rather, our empirical results suggest that the sufficient condition on $R$ may be not optimal. Thus, as we described in Limitations section (Section 7), improving theorem remains as future work. Even so, our theoretical results (especially the main inequality) provide meaningful insight into **why** the iterative randomization boosts the accuracy in pruning randomly-weighted neural networks: **the iterative randomization virtually increases the network width and thus its capacity**.
>
> ## References
>
> [1] V. Ramanujan et al. "What’s Hidden in a Randomly Weighted Neural Network?," CVPR 2020.

---

### Official Review · Reviewer_GSGE · 2021-07-17

**Rating:** 6
**Confidence:** 5

**Summary:**

This paper proposes utilizing the existing "lottery ticket" result with the re-initialization method. This work has some novelty since doing re-initialization is not studied in other papers.



**Limitations And Societal Impact:**

Yes

**Main Review:**

Questions: (1) The performance is actually no better than the previous method, for eg. ResNet18 on Cifar10 as compared to https://openreview.net/pdf?id=U_mat0b9iv. As a result, the performance gain by IteRand as compared to edge-popup is not clear. (2) For Theorem 4.1, the value of R is chosen unrealistically. Even for l=10, d=1000, R needs to be at least 10k/\epsilon. (3) The technical difficulty is not clear to me as compared to previous proofs. This should be emphasized. (4) What happens if the method is applied to NLP problems?

-----------------------After rebuttal----------------------------
The authors resolved my concern so that I increased my score.

**Time Spent Reviewing:**

1.5 hours

---

> ### Author Response · Authors · 2021-08-10
> **Replies for your valuable review**
>
>
> Thanks for commenting our paper.
>
> > (1) The performance is actually no better than the previous method, for eg. ResNet18 on Cifar10 as compared to https://openreview.net/pdf?id=U_mat0b9iv.
>
> The naive comparison between our method (**IteRand**) and the method in the cited paper (**biprop** [1]) is inappropriate. Biprop is designed to achieve SoTA of *binary neural networks* (BNNs). Although biprop uses **edge-popup** [2], which is the foundation of IteRand, they also leverage additional techniques to improve the accuracy of their BNN. For instance, much longer training epochs ($250$ epochs, more than twice as longer as [2] and our paper), training weights of batch normalizations ([2] and our paper use batch normalization without training), and techniques (e.g., learnable gain terms) from previous work on BNN. Thus, the naive comparison between IteRand and biprop in terms of reported accuracy does not provide any insight that IteRand is no better than edge-popup.
>
> Rather, we consider that **IteRand can improve the test accuracy of biprop** since they are orthogonal to each other. To verify this, we conducted additional experiments based on the codebase [3], which is the official repository of the biprop paper [1]. We implemented IteRand in this codebase, and compared the original biprop and biprop+IteRand (i.e. we replaced the edge-popup part with IteRand) with the same hyperparameters in their paper [1]. The results are the following:
>
> |  Networks \ Methods |   biprop | biprop+IteRand |
> | -------- | -------- | -------- |
> | ResNet-18 on CIFAR-10    | $93.89 \pm 0.13\\%$    | $\mathbf{94.25} \pm 0.18\\%$  |
>
> Here we show the mean ± standard deviation over three runs with different random seeds for each method. These results indicate that **the performance gain by IteRand is also effective in the context of BNN**. We consider the application of IteRand to studies on BNNs is an interesting direction of future research.
>
>
> > As a result, the performance gain by IteRand as compared to edge-popup is not clear.
>
> In our paper, to appropriately and fairly compare the baseline method (edge-popup [2]) and our method (IteRand), we trained both methods under the same condition until converged. Also we followed the training methods and hyperparameters in [2] as possible as we can. **Under these careful settings, Figure 2 and 3 in our paper clearly show the performance gain by IteRand in multiple experiments** (e.g., $+2.08\%$ in accuracy for ResNet-50 on ImageNet. Details are described in Section B2 in appendix.pdf, Supplementary Material).
>
> Also, the test accuracies achieved by IteRand actually outperform the reported accuracies of edge-popup in [2]. For example, IteRand achieves 69.19% for ResNet-50 and 72.99% for ResNet-101 on ImageNet, which outperform the reported accuracies of edge-popup, 68.6% for ResNet-50 and 72.3% for ResNet-101. Therefore, we conclude that the performance gain by IteRand is clear.
>
> NOTE: Ramanujan et al. [2] train these networks with full training dataset, whereas we split it into train/validation dataset and only use the former dataset (lines 233-234 in our paper). Thus, by nature, all test accuracies in our paper tend to be slightly degraded compared to [2] since our experiments were conducted with smaller training data.
>
> > (2) For Theorem 4.1, the value of R is chosen unrealistically. Even for l=10, d=1000, R needs to be at least 10k/\epsilon.
>
> We agree with this comment. As you pointed out, the sufficient condition on $R$ given in Theorem 4.1 becomes very large for deep and wide networks such as ResNets. However, this is just a sufficient condition but may not be necessary, and thus smaller $R$ can be possible. Indeed, in our experiments, IteRand works well with $R \approx 100$ for CIFAR-10 and $R \approx 1000$ for ImageNet, which is greatly smaller than the sufficient condition on $R$. As we mentioned in Limitation (Section 7), we believe that the sufficient condition in the theorem could be improved further in future work.
>
> Even so, our theoretical results have some other contributions. In particular, it provides meaningful insight into **why** the iterative randomization boosts the accuracy in pruning randomly-weighted neural networks: **the iterative randomization virtually increases the network width and thus its capacity**. Also, we emphasize that our formulation of pruning iteratively randomized neural networks **is novel in itself** and **generalizes** the formulation of all prior works [4-6].
>
> > (3) The technical difficulty is not clear to me as compared to previous proofs. This should be emphasized.
>
> We are sorry for the unclearness. As described in Section 4.3, the technical contribution in our theory is twofold:
> 1. **New formulation** (lines 154-160). We formulate pruning iteratively randomized neural networks as an approximation theorem involving $R\in\mathbb{N}$, the number of re-sampling operations. Our formulation generalizes the one in prior works [4-6] that consider only the case of neural networks with randomly-initialized and fixed weights. Also we introduced a projection map $\pi: \widetilde{I} \to I$ (Eq. 13) specific to our formulation and essentially used it to calculate probabilities (lines 217-219).
> 2. **New splitting identity** (lines 182-185). Let $f(x):=wx, (x,w\in\mathbb{R})$ be a linear function. All prior works [4-6] used the splitting identity $f(x) = \sigma(wx) - \sigma(-wx)$, where $\sigma$ is the ReLU activation, as an essential step in their proof. However, this splitting does not work well in our formulation. Indeed it only leads to the required width factor $2d\lceil O(... / R^2) \rceil$, where the factor never reduce to a constant even if $R$ is enough large, as opposed to our factor $2\lceil O(... / R^2) \rceil$ in Theorem 4.1. For this problem, our idea is to use an alternative splitting identity $f(x) = w\sigma(x) - w\sigma(-x)$, which finally enabled us to prove Theorem 4.1.
>
> We would like to emphasize these theoretical contributions in the final version.
>
> > (4) What happens if the method is applied to NLP problems?
>
> Although our experiments are focused on vision tasks, we expect that our method also works well on other tasks like NLP. Indeed, our method can be applied to any type of neural networks and tasks that edge-popup [2] can be applied. For example, edge-popup can be applied to NLP tasks with Transformer networks [7]. Also, our theoretical results indicate that IteRand may improve the parameter inefficiency of edge-popup in a task-agnostic way. Even so, since we agree that evaluating IteRand on different tasks like NLP is important, we would like to add some experimental results to the final version of our paper.
>
>
>
>
> ## References
>
> [1] J. Diffenderfer and B. Kailkhura. "Multi-Prize Lottery Ticket Hypothesis: Finding Accurate Binary Neural Networks by Pruning A Randomly Weighted Network," ICLR 2021.
>
> [2] V. Ramanujan et al. "What’s Hidden in a Randomly Weighted Neural Network?," CVPR 2020.
>
> [3] https://github.com/chrundle/biprop
>
> [4] E. Malach et al. "Proving the Lottery Ticket Hypothesis: Pruning is All You Need," ICML 2020."
>
> [5] L. Orseau et al. "Logarithmic Pruning is All You Need," NeurIPS 2020.
>
> [6] A. Pensia et al. "Optimal Lottery Tickets via SubsetSum: Logarithmic Over-Parameterization is Sufficient," NeurIPS 2020.
>
> [7] V. Sanh et al. "Movement Pruning: Adaptive Sparsity by Fine-Tuning," NeurIPS 2020.

---

### Author Response · Authors · 2021-08-10
**Summary of our responses**

We appreciate all reviewers for their careful reading and constructive feedback. We are glad the reviewers found our paper to be well-written and clearly motivated (Reviewer **5mau**, **HPM9**, **x4sx**), our method to be novel (**5mau**, **HPM9**, **x4sx**, **GSGE**), very intuitive (**x4sx**) and interesting (**5mau**).  Also, three reviewers (**5mau**, **HPM9**, **x4sx**) agree on the empirical gain by our method.

There are two shared concerns about our theoretical and empirical results:

1. The sufficient condition for the number of re-samplings $R$, which is a corollary of the main inequality (Eq. 7), is unrealistically huge. (Reviewer **x4sx**, **GSGE**)
2. How much does our method **IteRand** have additional computational cost over **edge-popup** [1] ? (Reviewer **HPM9**, **x4sx**)

For these concern and question, our answers can be summarized as follows:

1. We agree that the sufficient condition for $R$ is relatively large, compared to our empirical results. Indeed, we used smaller $R$ in our experiments and it works well: $R \approx 100$ on CIFAR-10 and $R \approx 1K$ on ImageNet. This gap indicates that there is still room for improving our theorem, as we described in Limitation section. Even so, our theoretical results (especially the main inequality) provide meaningful insight into **why** the iterative randomization boosts the accuracy in pruning randomly-weighted neural networks: **the iterative randomization virtually increases the network width and thus its capacity**. Moreover, we emphasize that our formulation of pruning iteratively randomized neural networks **is novel in itself** and **generalizes** the formulation of all prior works.
2. The additional computational cost is **negligibly small** in all our experiments. We measured the average cost of a single randomizing operation, which is the only difference from edge-popup, as follows: 97.10 ms for ResNet-18 (11.2M params), 200.55 ms for ResNet-50 (23.5M params). Thus, **the total additional cost should be about 10 seconds in the whole training ($\approx$ 1.5 hours) for ResNet18 on CIFAR-10 and 200~300 seconds in the whole training ($\approx$ one week) for ResNet50 on ImageNet**. These estimates are consistent with the actual training time. We will add these analysis to the final version of our paper.

Moreover, we obtained additional empirical results during the process of answering other questions and comments:

- We found that, **for ResNet-18 on CIFAR-10, edge-popup requires at least 1.3 times wider networks than IteRand**. This result makes the performance gain by IteRand more clear. (in the response to Reviewer **HPM9**)
- We confirmed that **IteRand consistently boosts the test accuracies of edge-popup with various pruning rates**. (in the response to Reviewer **HPM9**)

Also, we answered the other questions and comments in the replies for each reviewer.

## References

[1] V. Ramanujan et al. "What’s Hidden in a Randomly Weighted Neural Network?," CVPR 2020.

---

### Decision · Program_Chairs · 2021-09-27

**Decision:**

Accept (Spotlight)

**Comment:**

The paper presents a new algorithm for learning by pruning. In the original work by Ramanujan et al. the random weights of a network could be pruned away so that the resulting network could achieve arbitrary accuracy for a given task, at the cost of slight overparameterization. The authors in the current paper show that the performance of such learning by pruning schemes can be improved by further rerandomizing the weights at consequent pruning iterations. This results in a reduced overaparemeterization to achieve the same accuracy compared to past results. The results are novel, and interesting with regards to the related literature. The reviewers have posted several points that the work can be improved, and all are addressed/can be addressed by the final version of the paper.